# Rethinking the Starting Point: Enhancing Performance and Fairness of Federated Learning via Collaborative Pre-Training

## Abstract

Most existing federated learning (FL) methodologies have been developed starting from a randomly initialized model. Recently, several studies have empirically demonstrated that leveraging a pre-trained model can offer advantageous initializations for FL. In this paper, we take a departure from the assumption of centralized pre-training and instead focus on a practical FL setting, where data samples are distributed among both clients and the server even during the pre-training phase. We propose a collaborative pre-training approach for FL (CoPreFL), where the goal is to strategically design a pre-trained model that effectively serves as a good initialization for any downstream FL tasks. The key idea of our pre-training algorithm is to employ meta-learning to simulate downstream distributed scenarios, enabling it to adapt to unforeseen FL tasks. During optimization, CoPreFL also strikes a balance between average performance and fairness, with the aim of addressing the challenges in downstream FL tasks through initialization. Extensive experimental results validate that our pre-training method provides a robust initialization for any unseen downstream FL tasks, resulting in enhanced average performance and more equitable predictions. *The code is also submitted.*

## 1 Introduction

Federated Learning (FL) has emerged as a popular distributed machine learning paradigm that facilitates collaborative model training among a set of clients through periodic aggregations of local models by a server (McMahan et al., 2017; Konecný et al., 2016). The inherent FL property of keeping data local to clients offers significant privacy advantages, making it highly appealing for numerous learning applications. Federated averaging (FedAvg), as the first FL technique, stands out as arguably the most widely used FL algorithm. Various other FL methodologies have been also proposed in the literature, such as aggregation schemes (Ji et al., 2019; Wang et al., 2020) and improved local training techniques (Reddi et al., 2021; Sahu et al., 2020). However, unlike the common practice of transfer learning in natural language processing (Radford et al., 2019; Devlin et al., 2019) and computer vision (Dosovitskiy et al., 2021), which typically involve learning from pre-trained models in a centralized setup, relatively few prior works have delved into the analysis of *model initialization* specifically for FL. Instead of starting from a well pre-trained model, most of the FL works initialize their models using random model weights.

**Background and observations.** Nguyen et al. (2023); Chen et al. (2023) were the first to systematically demonstrate that initializing FL with centrally pre-trained models can improve the FL performance. However, they assume that the pre-training dataset is centrally stored, which might often not align with practical scenarios where data is distributed across multiple sources. Consequently, their initialization approach may not offer a viable solution when dealing with pre-training in distributed settings. Moreover, while the centrally pre-trained model does lead to performance improvements in downstream FL, they tend to overlook potential side effects, such as performance biases, that may emerge in downstream FL tasks when utilizing these models. As illustrated by the histograms in Figure 1, although utilizing the centrally pre-trained model as a FL initialization enhances performance compared to random initialization, it introduces substantial performance variance across clients in downstream FL tasks. This phenomenon may arises from the lack of generalizability in the model's design. When a model pre-trains in a centralized manner, it becomes rigidly bound to

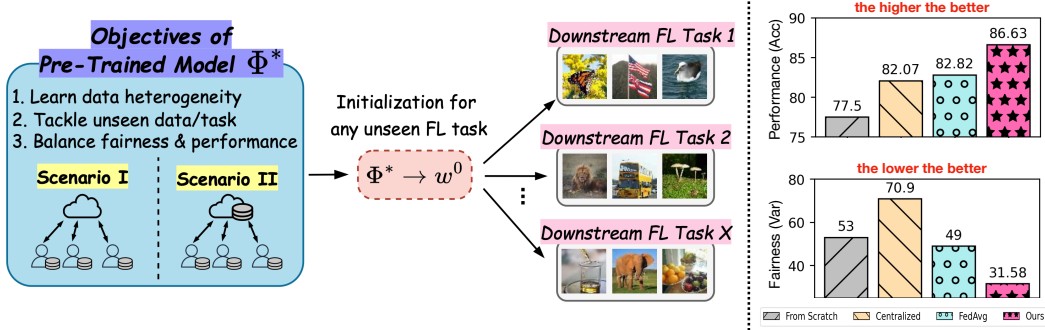

Figure 1: (Left): Overview of `CoPreFL`, aiming to provide a robust initialization for any unseen downstream FL task. (Right): Average accuracy and variance of FL initialized by various pre-trained models. `CoPreFL` demonstrates improved FL performance in terms of both accuracy and fairness.

the knowledge in the pre-training dataset. This fixity poses a challenge in adapting the model to the diverse clients that may contain new or unseen data in downstream tasks. This variance can result in fairness issues, as achieving balanced predictions among FL clients becomes challenging (Li et al., 2020; Cho et al., 2022).

**Challenges and goals.** From the service provider's perspective, we are interested in initializing FL models via pre-training in a distributed setup as shown in the overview of Figure 1. By taking advantage of currently available clients, the goal is to construct a robust pre-trained model that can serve as a robust initialization for future downstream FL tasks. This problem poses several challenges. Firstly, the initialization must enhance performance without introducing large accuracy variance among clients in the downstream FL task, to address fairness concerns. Secondly, the initialization should effectively handle unseen data and labels since data for pre-training and downstream FL tasks are usually disjoint due to the time-varying environment or new clients joining the system, e.g., classifying different objects depending on the self-driving car's environment or face/speech recognition for new phone users. This necessitates an initialization that must generalize well to unfamiliar data, accommodating shifts in the task domain or objectives. Lastly, an ideal initialization should be versatile, offering a solid starting point not only for just one FL task but for any FL tasks, ensuring its applicability across various scenarios in FL. Addressing these challenges in a distributed pre-training scenario is becoming increasingly important and aligns with many practical settings, as the dataset in the pre-training phase in not always centralized. Considering that the downstream tasks also involve FL, a FL-based pre-training approach might closely resemble the downstream scenario, potentially providing an informative initialization that resonates with the nature of FL. This naturally leads to the question: *How can we design a FL-based pre-training methodology that effectively addresses the challenges that are encountered in downstream FL tasks?*

**Contributions.** We propose `CoPreFL`, a collaborative FL-based pre-training approach that provides a good initialization for any downstream FL, by handling the aforementioned challenges through meta-learning to ensure reliable predictions even for unseen downstream FL tasks. Our pre-training approach begins by replicating the real-world scenarios faced in downstream FL, leveraging pre-training data in two distributed scenarios: one where data is solely gathered from distributed clients and another where, in addition to clients holding data, the server also possesses a portion of the dataset. We employ model-agnostic meta-learning (MAML) to enable the pre-trained model to handle challenges related to unseen data by adapting the model using meta-updates, thereby addressing the issue of disjoint data between pre-training and downstream FL. Moreover, we optimize the pre-trained model not solely based on the average performance of participants during the pre-training phase, but also by balancing the objective through variance across participants. This comprehensive approach aims to alleviate the fairness concerns in downstream FL, ensuring both improved average performance and fairer predictions across clients. In developing `CoPreFL`, our contributions are as follows:

- We systematically analyze initialization for FL, by relaxing the assumption of pre-training data being stored centrally. We specifically demonstrate the viability of pre-training in a hybrid distributed server-client data storage setting, encompassing scenarios where either only distributed clients hold data or where the server also retains a portion of the data.

- Our method employs meta-learning to replicate distributed scenarios and optimize the pre-trained model for both average accuracy and fairness, effectively providing a robust initialization that implicitly addresses challenges in any unforeseen downstream FL tasks.

- Extensive experiments reveal that `CoPreFL` consistently offers remarkable improvements in both average performance and fairness of predictions across clients for any downstream FL task, compared with various centralized/distributed pre-training baselines.

Finally, we would like to highlight that several existing works adopt meta-learning during pre-training in a FL setup, to construct an initial model for personalization (Jiang et al., 2019; Fallah et al., 2020; Collins et al., 2021). However, the focus of their downstream tasks are client-side local learning, not FL, which leads to limited performance when clients in the downstream tasks aim to collaboratively train a global model via FL, as we will see in Section 4. To the best of our knowledge, this is one of the earliest works that consider FL in both pre-training and downstream stages, with several unique characteristics including hybrid client-server learning and balancing between average performance and fairness during pre-training.

## 2    RELATED WORK

**Pre-training for FL.** While pre-training has been extensively applied in AI applications for the centralized setup (Radford et al., 2019; Brown et al., 2020; Devlin et al., 2019; Dosovitskiy et al., 2021; Kolesnikov et al., 2019), its potential effects on FL have remained relatively unexplored. Several recent works have studied the effect of model initialization in FL Nguyen et al. (2023); Chen et al. (2023); Stremmel & Singh (2021) by systematically comparing randomly initialized model with the pre-trained model. It is shown that conducting FL starting from a well pre-trained model can significantly enhance the model performance. However, they consider pre-training in a centralized setup where all pre-training samples are available at the central server, which may not hold in many practical settings where data samples are distributed across the clients even during the pre-training stage. Moreover, centrally pre-trained models introduce further variance challenges in downstream FL tasks, as observed in Figure 1. Compared to these works, we develop a new *pre-training strategy tailored to distributed settings*, so that initializing FL with the pre-trained model can address the challenges on unseen data and performance fairness in any downstream FL tasks.

**Meta-learning in FL.** Our federated pre-training approach employs meta-learning techniques to adapt the model toward our controlled objectives, providing any FL task with a strong initialization. This distinguishes it from other FL methods that also frequently utilize meta-learning, such as personalized FL and few-round FL. Park et al. (2021) introduced few-round FL, which employs a meta-learning-based episodic training strategy to adapt FL to any group within only a few rounds of FL. However, in (Park et al., 2021), fairness is not considered in downstream FL tasks, and the solution for a practical scenario in which the server holds data is not provided. In personalized FL (Chen et al., 2018; Jiang et al., 2019; Fallah et al., 2020; Collins et al., 2021) (e.g., FedMeta (Chen et al., 2018)), a global model is formed through aggregating local updates and meta-updates at each federated participant. The obtained global model serves as an initialization at each client to achieve a personalized local model through a limited number of gradient descent steps. Unlike this research direction where the downstream tasks are personalization to individual clients, our emphasis is on establishing an initial model that results in a high-accuracy and equitable global model for *any FL downstream tasks*. This makes our objective function and meta-learning strategy different compared to existing works, leading to significant performance improvements as we will observe in Section 4.

**Performance fairness in FL.** Performance fairness has been studied in the FL literature (Mohri et al., 2019; Li et al., 2020; Cho et al., 2022) aiming to construct a global model that satisfies as many clients as possible (e.g., a global model that achieves uniform distribution of accuracy across participants). Constructing a fair global model is important as the fair model is more likely to satisfy the new clients joining the FL system, without additional model training. The primary objective of our pre-training method is to construct a *robust initial model* that guarantees performance for all participants and enhance performance fairness, for any downstream FL scenarios.

## 3    PROPOSED PRE-TRAINING

In this section, we begin by establishing the classical objective of FL and outlining the scenarios during the pre-training phase. We then present our pre-training objectives, and propose `CoPreFL`, a meta-learning based pre-training approach which simulates the conditions that could be encountered when applying a pre-trained model to the downstream FL task.

### 3.1 PROBLEM SETUP AND OBJECTIVES

**Federated learning (downstream task).** Consider a FL task where a central server is connected to a set of clients $G$. Starting from the initialized model $w^0$, FL consists of parallel local training at the clients and global aggregation at the server across multiple communication rounds. In each communication round $r$, every client $g \in G$ downloads the previous global model $w^{r-1}$ from the server and subsequently updates it using their local dataset $D_g$. The goal of this local training is to maximize each client's gain, often achieved through multiple iterations of stochastic gradient descent (SGD). After all clients have completed their local model updates, the updated local models, denoted as $w_g^r$, are uploaded to the server for aggregation. This aggregation results in a new global model $w^r = \sum_{g \in G} \frac{|D_g|}{|D|} w_g^r$ by employing FedAvg (McMahan et al., 2017), where $D$ represents the aggregated training set comprising data from all clients and $|D|$ denotes the number of samples in dataset $D$. This entire process is iterated for $R$ communication rounds until convergence.

**Pre-training scenarios.** Existing FL approaches primarily rely on randomly initialized models, while some recent studies have explored the potential of a centralized pre-training setup (Nguyen et al., 2023; Chen et al., 2023). In this paper, we depart from the assumption of centralized pre-training data, and consider a practical yet challenging scenario in which data samples are distributed across the clients even during the pre-training stage. Here, the labels and data that appear in the pre-training stage are potentially different from the ones in the downstream tasks. Our objective is to design a collaborative pre-training strategy tailored to this new problem setup. We explore two distinct FL scenarios in our analysis, as depicted in the overview figure in Figure 1. In addition to the conventional FL scenario where datasets are exclusively available at the clients **(Scenario I)**, we also considered a hybrid FL scenario where the server holds a small amount of data **(Scenario II)**. In various real-world scenarios, the entity in charge of creating the machine learning model manages the server and holds a limited dataset that approximates the broader population distribution. As a result, a hybrid FL strategy that combines the abundant client data with a small portion of server data in a decentralized and privacy-preserving fashion is becoming imperative to substantially boost model performance in real-world applications (Yang et al., 2023; Bian et al., 2023).

**`CoPreFL` objectives.** Instead of randomly initializing the starting model for downstream FL task, our objective is to design a pre-trained model $\Phi^*$ under both scenarios I and II, that serves as a robust starting point $w^0 = \Phi^*$ for any unseen downstream FL task, as shown in Figure 1. More precisely, one of the goals of $\Phi^*$ is to minimize the following objective function for downstream FL tasks:

$$A(\Phi) = \mathbb{E}_{G \sim p(\mathcal{G})} \left[ \frac{1}{|G|} \sum_{g \in G} f(w^R(\Phi^*, G), D_g) \right], \tag{1}$$

where $p(\mathcal{G})$ represents the distribution of all possible sets of clients for downstream FL, $G$ stands for a specific group of clients drawn from $p(\mathcal{G})$, $f(\cdot)$ represents the loss function, $w^R(\Phi^*, G)$ symbolizes the final $R$-th round global model derived from clients in set $G$ starting with $\Phi^*$ as initialization, and $D_g$ represents the downstream dataset of client $g$ within client set $G$. The metric in (1) represents the average FL performance across all clients that could possibly appear in the downstream tasks.

On the other hand, FL can lead to significant variations in performance among different participants, particularly when the model exhibits bias towards those with larger datasets, emphasizing the importance of *fairness* in performance distribution across all participants. This fairness can be assessed by quantifying the variance in testing accuracy across participants (Li et al., 2020). Therefore, in addition to achieving performance gains on any unseen FL task, we also aim for the final global model $w^R(\Phi^*, G)$ initialized from our designed pre-trained model $\Phi^*$ to exhibit a fair testing performance distribution across $|G|$ clients. To be more specific, the second goal of $\Phi^*$ is to minimize the variance of the prediction distribution across participants in downstream FL tasks:

$$F(\Phi) = \mathbb{E}_{G \sim p(\mathcal{G})} \left[ \frac{1}{|G|} \sum_{g \in G} f^2(w^R(\Phi^*, G), D_g) - \left( \frac{1}{|G|} \sum_{g \in G} f(w^R(\Phi^*, G), D_g) \right)^2 \right]. \tag{2}$$

We aim to minimize and strike a balance between (1) and (2); however, we encounter challenges because $D_g$, $G$, and $p(\mathcal{G})$ are not known during pre-training, preventing us from directly optimizing (1) and (2). Our idea is to employ meta-learning during pre-training to simulate downstream FL

---

**Algorithm 1** Our Pre-training Method `CoPreFL` (Pre-training Phase in Scenario I)

---

1: **Input:** A set of clients $M$ in the pre-training phase, with each client $i$ holding its dataset $D_i^p$.
2: **for** Each communication round $t = 1, 2, ..., T$ **do**
3:     Randomly select a set of clients $m \subset M$ to participate in learning
4:     Each participant $j \in m$ partitions its own dataset $D_j^p$ into support set $S_j$ and query set $Q_j$
5:     **for** Each participant $j$ in parallel **do**
6:         Downloads $\Phi^{t-1}$ from the server
7:         $\mathcal{L}_{S_j}(\Phi^t) \leftarrow \frac{1}{|S_j|} \sum_{(x,y) \in S_j} \ell(\Phi^{t-1}(x), y)$          ▷ Compute local loss using support set $S_j$
8:         $\Phi_j^t \leftarrow \Phi^{t-1} - \eta \nabla \mathcal{L}_{S_j}(\Phi^t)$          ▷ Perform SGD local update using support loss $\mathcal{L}_{S_j}$
9:     **end for**
10:     $\overline{\Phi^t} \leftarrow \sum_{j \in m} \frac{|S_j|}{\sum_{i \in m} |S_i|} \Phi_j^t$          ▷ Model aggregation to construct temporary global model
11:     **for** Each participant $j$ in parallel **do**
12:         Participant downloads $\overline{\Phi^t}$ for initialization and performs meta-updates
13:         $\mathcal{L}_{Q_j}(\overline{\Phi^t}) \leftarrow \frac{1}{|Q_j|} \sum_{(x,y) \in Q_j} \ell(\overline{\Phi^t}(x), y)$          ▷ Compute local meta-loss using query set $Q_j$
14:     **end for**
15:     Overall query meta-loss: $\mathcal{L}_Q(\overline{\Phi^t}) = \sum_{j \in m} \mathcal{L}_{Q_j}(\overline{\Phi^t})$; Variance across meta-losses: $\sigma_Q^2(\overline{\Phi^t})$
16:     Customized query meta-loss: $\mathcal{L}_{meta}(\overline{\Phi^t}) = \gamma \mathcal{L}_Q(\overline{\Phi^t}) + (1 - \gamma)\sigma_Q^2(\overline{\Phi^t})$
17:     $\Phi^t \leftarrow \overline{\Phi^t} - \zeta \nabla \mathcal{L}_{meta}(\overline{\Phi^t})$          ▷ Model updates using customized loss
18: **end for**
19: **Output:** A pre-trained model for downstream FL tasks: $\Phi^T$

---

scenarios and learn a pre-trained model capable of providing robust initialization for any unseen downstream FL task, considering (1) and (2).

**CoPreFL overview.** To create a pre-trained model that is well-suited for the diverse downstream FL tasks, we construct a pre-training environment that also mirrors the federated setup, facilitating the pre-trained model's ability to learn data heterogeneity. Our meta-learning-based `CoPreFL` involves iteratively learning of the pre-trained model over federated rounds using the support set, followed by a concluding adjustment (meta-update) using the query set. By treating the query set as unseen knowledge, we equip our pre-trained model with the capability to effectively handle unforeseen FL scenarios in downstream tasks and address our two goals (1) and (2).

### 3.2 CoPreFL in Scenario I (Pre-training with Distributed Clients)

We first consider a scenario where pre-training data is collected from distributed clients, and no data is stored on the server. The key challenge we address is that the data and FL scenarios in downstream tasks are inherently unseen during the pre-training phase. The detailed procedure of `CoPreFL` is given in Algorithm 1. We first randomly involve a set of clients $m$ to participate in each round, where each client $j$ holds its own dataset $D_j^p$. Prior to starting our `CoPreFL`, each participant $j$ splits its local dataset into support set $S_j$ and query set $Q_j$, which are disjoint. We apply meta-learning based on support and query sets to maximize the model's generalization ability in unseen scenarios.

**Temporary global model construction.** In each FL round $t$ in the pre-training phase, participants download model $\Phi^{t-1}$ from the server (line 6 in Algorithm 1). Subsequently, clients engage in a series of local training iterations using their respective support sets $S_j$ (line 7 in Algorithm 1), resulting in a training support loss $\mathcal{L}_{S_j}(\Phi^t) = \frac{1}{|S_j|} \sum_{(x,y) \in S_j} \ell(\Phi^{t-1}(x), y)$, where $\ell(\cdot)$ denotes the loss function (e.g., cross-entropy loss). Participants then update their local models using the loss $\mathcal{L}_{S_j}(\Phi^t)$, resulting in their respective updated models, denoted as $\Phi_j^t$ (line 8 in Algorithm 1). After all participants have completed their local training, they upload their models to the server (line 10 in Algorithm 1), which are aggregated according to $\overline{\Phi^t} = \sum_{j \in m} \frac{|S_j|}{\sum_{i \in m} |S_i|} \Phi_j^t$. This model can be viewed as the temporary global model that will be further updated using the query sets.

**Average performance and fairness.** Firstly, the query sets are used to evaluate the model's performance on each client and to conduct the meta-update process. The objective of our pre-trained model is to strike a balance between the following functions during the pre-training phase:

$$\min_{\Phi} \mathcal{L}_Q(\overline{\Phi^t}) = \min_{\Phi} \sum_{j \in m} \mathcal{L}_{Q_j}(\overline{\Phi^t}) \text{ and} \qquad (3)$$

$$\min_{\Phi} \sigma_Q^2(\overline{\Phi^t}) = \min_{\Phi} \frac{1}{|m|} \sum_{j \in m} \left( \mathcal{L}_{Q_j}(\overline{\Phi^t}) - \frac{1}{|m|} \mathcal{L}_Q(\overline{\Phi^t}) \right)^2, \tag{4}$$

where $\mathcal{L}_{Q_j}$ represents the meta-loss evaluated using the query set $Q_j$ of participant $j$, $\mathcal{L}_Q$ denotes the overall query meta-loss, which is characterized by aggregating $\mathcal{L}_{Q_j}$ across all participants, and $\sigma_Q^2$ represents the performance variance evaluated using the query set across participants. Beyond merely equipping the global model with the ability to achieve a good average performance in objective (3), we also strive to optimize the model for uniform/fair prediction performance across all participants in objective (4). Specifically, we tailor a customized query meta-loss function $\mathcal{L}_{meta}(\overline{\Phi^t})$ to minimize not only the overall query meta-loss $\mathcal{L}_Q(\overline{\Phi^t})$ when encountering unseen data but also the variance $\sigma_Q^2(\overline{\Phi^t})$ of query meta-losses across participants:

$$\min_{\Phi} \mathcal{L}_{meta}(\overline{\Phi^t}) = \min_{\Phi} \left[ \gamma \mathcal{L}_Q(\overline{\Phi^t}) + (1 - \gamma) \sigma_Q^2(\overline{\Phi^t}) \right], \tag{5}$$

where $\gamma \in [0, 1]$ represents the balancer acting as a control to strike a balance between these two factors effectively. Setting $\gamma = 0$ encourages a more uniform training accuracy distribution and improves fairness, aligning with objective function (4), but it may sacrifice performance. A larger $\gamma$ means that we emphasize the devices' average performance with less consideration for uniformity, optimizing the pre-trained model more towards objective function (3).

**Meta update.** Considering the above objective function, each participant $j$ downloads the temporary global model $\overline{\Phi^t}$ and employs its query set $Q_j$ to compute and local query loss $\mathcal{L}_{Q_j}(\overline{\Phi^t}) = \frac{1}{|Q_j|} \sum_{(x,y) \in Q_j} \ell(\overline{\Phi^t}(x), y)$ as in line 13 in Algorithm 1; the gradients are computed locally and set back to server for aggregation for later use. Subsequently, the overall query meta-loss $\mathcal{L}_Q(\overline{\Phi^t})$ is computed by aggregating all local query losses, and the performance variance $\sigma_Q^2(\overline{\Phi^t})$ is determined by analyzing query meta-losses across all participants (line 15 in Algorithm 1). Then, as described in line 17 of Algorithm 1, we update the temporary global model $\overline{\Phi^t}$ using the customized query meta-loss $\mathcal{L}_{meta}$ and the aggregated received gradients to align with our controlled objective. The server finally broadcasts the meta-updated global model $\Phi^t$ to the participants and proceeds to the next round. After $T$ federated rounds, the final global model $\Phi^T$ serves as the pre-trained model for initializing FL in the downstream tasks: As illustrated in Figure 1, the set of clients in any downstream tasks conduct FL starting from the pre-trained model $w^0 = \Phi^T$.

## 3.3 CoPreFL in Scenario II (Hybrid Client-Server Pre-Training)

In scenario II, in addition to distributed clients holding the data, we explore a pre-training scenario where the server also possesses a small portion of data approximating the broader population distribution. Similar to CoPreFL in scenario I, our primary aim remains optimizing two objective functions (3) and (4) during pre-training phase to achieve the goals (1) and (2). The key distinction is that we achieve these objectives through meta-updates performed on the server's data, rather than on the participants' data. Unlike CoPreFL in scenario I where we separated participants' data into support and query sets, viewing the query set as unseen knowledge to control average performance and fairness, in scenario II, we employ every data sample within each participant for local updates by viewing as support data. Simultaneously, we treat the server's data as unseen knowledge, i.e., query set, allowing us to customize the model according to our objectives.

The detailed procedure of CoPreFL in scenario II is given in Algorithm 2 in Appendix A. The goal remains to balance between objective functions (3) and (4), with the first key difference compared to CoPreFL in scenario I being that the temporary global model $\overline{\Phi^t}$ is aggregated from local models, each of which learns from its entire local dataset $D_j^p$ instead of their support sets. The second key difference is that we facilitate the meta-update of the temporary global model $\overline{\Phi^t}$ using server's data instead of client's data. Specifically, we randomly partition the server's dataset $D^s$ into $|m|$ equal partitions, emphasizing that this partitioning is not obligatory for the server but is undertaken to mimic the distributed nature of the scheme and furnish distributed participants with a global model suited to our two objectives (3) and (4). We then strike a balance between optimizing for performance and fairness by updating the temporary global model $\overline{\Phi^t}$ based on a customized server meta-loss $\mathcal{L}_{meta}(\overline{\Phi^t})$, calculated through meta-updates on the server's partitioned data.

**Remark 1.** We note that our meta update is applied to the *temporary global model* to mimic downstream FL scenarios, which is basically different from existing meta-learning based FL methods

that meta update the local models to mimic local/personalized training. This leads to significant performance improvement of CoPreFL compared with the baseline, as we will see in Section 4.

**Remark 2.** Although we described CoPreFL in two different distributed scenarios, CoPreFL is applicable even when all data samples are centralized at the server during pre-training, to construct a more robust initial model compared with naive pre-training. The server can split the dataset to mimic either scenarios I or II, and directly apply the above training strategies.

## 4 EXPERIMENTS

### 4.1 EXPERIMENTAL SETUP

**Dataset and model.** We evaluate the performance of our algorithm on CIFAR-100 (Krizhevsky, 2009), Tiny-ImageNet (Le & Yang, 2015), FEMNIST (Caldas et al., 2018) datasets, adhering to the data splits provided in (Ravi & Larochelle, 2016; Park et al., 2021). We perform image classification on the mentioned datasets using the ResNet-18 model (He et al., 2015). See Appendix B.1 for detailed settings.

**Pre-training phase.** We distribute the pre-training dataset to $|M| = 100$ clients following either IID or non-IID data distributions and select $|m| = 20$ participants out of the $|M|$ clients to participate in each FL round. Results with different $|m|$ are reported in Appendix C. Each participant employs 80% of its local data as support samples and the remaining 20% as query samples. We set the number of global rounds to $T = 50$, and each round of local training takes 5 iterations for each participant. See Appendix B.2 for detailed hyperparameters and compute settings.

**Downstream FL task and evaluation metrics.** As illustrated in Figure 1, the final global model of the pre-training phase is utilized as initializing each downstream FL task, where we consider multiple downstream tasks to evaluate the overall performance. To generate each downstream FL task, we randomly select 5 classes from each downstream datasets and fix the training procedure at each downstream task to the commonly adopted FedAvg algorithm. We evaluate the final global model's performance using testing samples from each participant and report both the accuracy and the variance of the accuracy distribution across participating clients for each FL task. Additionally, we present the worst 10%, 20%, and 30% testing accuracies among all clients for each FL task, aiming to evaluate the capability of each method in handling underperforming participants. The detailed configurations for downstream FL task can be found in Appendix B.3.

**Data distribution.** In the IID setup, data samples from each class are distributed equally to $|M| = 100$ clients for pre-training and $|G| = 10$ clients for downstream FL task. For the non-IID setup, samples within each class are partitioned among $|M|$ and $|G|$ clients using a Dirichlet($\alpha$) distribution for pre-training and downstream task, respectively, with $\alpha = 0.5$ selected as is in the literature (Morafah et al., 2022; Li et al., 2021).

**Baselines for pre-training.** We compare CoPreFL with various established FL algorithms, including standard FedAvg (McMahan et al., 2017), FedMeta (Chen et al., 2018) which addresses the unseen scenario through meta-learning, and q-FFL(q > 0) (Li et al., 2020) which aims at enhancing performance fairness across clients. All of these schemes are adopted during the pre-training phase to construct initial models. Across all FL algorithms, we maintain consistent settings for the number of global rounds, local iterations, the chosen $|m|$ participants, optimizer, and local learning rate. When applying these baselines in **scenario II**, after each method completes its local iterations and obtains its respective global model in each round, we proceed to further train the global model with 5 additional iterations, utilizing the server dataset. This extended training, optimized using the SGD optimizer with a learning rate of $10^{-3}$, follows the hybrid training approach introduced in Yang et al. (2023); Bian et al. (2023), where the server's data is used to further refine the global model. Subsequently, the server broadcasts the server-trained global model to each participant for the next round. Similarly, we introduce a baseline that constructs the global model according to Algorithm 1 and then further performs multiple SGD iterations using server data. For differentiation, we denote this baseline as "CoPreFL-SGD", which is evaluated only for scenario II. Finally, we also see the effects of random initialization and conventional centralized pre-training, in Table 5.

### 4.2 EXPERIMENTAL RESULTS

**Results for scenario I.** Tables 1 and 2 show test accuracies averaged over 10 different non-IID FL downstream tasks, initialized with various pre-trained methods in scenario I on CIFAR-100 and

| Pre-training (Scenario I) | | Downstream: Non-IID FL | | | | |
|---|---|---|---|---|---|---|
| Distribution | Method | Acc ↑ | Variance ↓ | Worst 10% ↑ | Worst 20% ↑ | Worst 30% ↑ |
| IID | FedAvg | 84.20 | 57.15 | 68.43 | 71.83 | 74.38 |
| | FedMeta | 83.80 | 42.64 | 72.30 | 73.79 | 75.47 |
| | q-FFL | 82.60 | 45.56 | 70.46 | 73.41 | 75.14 |
| | CoPreFL ($\gamma = 0.25$) | **84.36** | **38.56** | **73.66** | **75.63** | **77.40** |
| Non-IID | FedAvg | 78.96 | 64.80 | 62.70 | 67.00 | 69.80 |
| | FedMeta | 82.45 | 48.72 | 68.97 | 72.41 | 74.35 |
| | q-FFL | 80.01 | 88.92 | 64.39 | 67.48 | 70.30 |
| | CoPreFL ($\gamma = 0.75$) | **83.29** | **34.69** | **71.58** | **73.20** | **74.59** |

Table 1: **Scenario I (CIFAR-100):** Average performance across 10 non-IID downstream FL tasks, initialized with various FL pre-trained models.

| Pre-training (Scenario I) | | Downstream: Non-IID FL | | | | |
|---|---|---|---|---|---|---|
| Distribution | Method | Acc ↑ | Variance ↓ | Worst 10% ↑ | Worst 20% ↑ | Worst 30% ↑ |
| IID | FedAvg | 79.45 | 35.40 | 64.86 | 68.61 | 70.78 |
| | FedMeta | 81.68 | 65.61 | 65.96 | 69.17 | 71.59 |
| | q-FFL | 82.65 | 39.69 | 70.65 | 74.14 | 76.27 |
| | CoPreFL ($\gamma = 0.0$) | **83.79** | **34.93** | **72.59** | **75.05** | **76.76** |
| Non-IID | FedAvg | 82.94 | 37.21 | 68.99 | 72.29 | 74.40 |
| | FedMeta | 81.03 | 37.58 | 69.44 | 71.55 | 72.93 |
| | q-FFL | 84.11 | 43.96 | 73.87 | 76.05 | 77.37 |
| | CoPreFL ($\gamma = 0.5$) | **85.23** | **35.40** | **76.77** | **78.46** | **79.86** |

Table 2: **Scenario I (Tiny-ImageNet):** Average performance across 10 non-IID downstream FL tasks, initialized with various FL pre-trained model.

Tiny-ImageNet datasets. Our pre-trained method `CoPreFL` stands out by offering a robust initialization that is characterized by higher average accuracy and reduced variance across clients in downstream FL tasks. Moreover, our method not only improves average performance and reduces variance across all clients but also increases accuracies for the worst-performing clients, as indicated by the Worst 10-30% metrics. This further highlights the advantage of balancing the two objective functions (3) & (4), particularly in benefiting the worst-performing clients. In Appendix C.1, we provide additional results considering varying the number of participants during pre-training scenario I and different data distributions in downstream FL tasks.

**Results for scenario II.** Table 3 displays the performance of various pre-trained methods trained in scenario II. Our `CoPreFL` outperforms other baselines, demonstrating that utilizing server data for meta-updates and striking a balance between objectives (3) and (4) with the server's data can still effectively align with goals (1) and (2). We observe that `CoPreFL` consistently outperforms `CoPreFL-SGD`, indicating that conducting a few SGD iterations using server data in a centralized manner after meta-updating the global model might diminish the effectiveness of our designed objectives. This further emphasizes the importance of performing meta-learning on server data, following the `CoPreFL` in Algorithm 2 for scenario II, to effectively address objective functions (3) and (4). See Appendix C.2 for additional results considering different configurations in scenario II.

**Fairness of `CoPreFL`.** Figure 2 depicts the testing accuracy distributions of the final global model on each client for 10 non-IID FL downstream tasks. We provide visualizations for our method, as well as the methods with the second-best average accuracy and the second-lowest variance. Our `CoPreFL` approach excels not only in achieving higher average accuracy compared to other pre-training methods but also in establishing more centered (i.e., fairer) testing accuracy distributions with reduced variance. Moreover, when considering the clients located on the left end of the distribution in each pre-training method, our approach effectively shifts low-performing clients towards right, signifying an enhancement in predictive accuracy for these clients. In Appendix C.3, we provide additional comprehensive visual results across different scenarios and configurations.

| Pre-training (Scenario II) | | Downstream: Non-IID FL | | | | |
|---|---|---|---|---|---|---|
| Distribution | Method | Acc ↑ | Variance ↓ | Worst 10% ↑ | Worst 20% ↑ | Worst 30% ↑ |
| IID | FedAvg | 81.79 | 41.73 | 69.84 | 73.47 | 75.11 |
| | FedMeta | 82.29 | 47.75 | 71.69 | 74.17 | 75.71 |
| | q-FFL | 82.40 | 40.32 | 73.96 | 75.30 | 76.59 |
| | CoPreFL-SGD ($\gamma = 0.75$) | 82.90 | 38.94 | 73.02 | 75.60 | 77.18 |
| | CoPreFL ($\gamma = 0.75$) | **85.68** | **27.14** | **75.36** | **77.25** | **78.49** |
| Non-IID | FedAvg | 82.82 | 49.00 | 69.71 | 72.54 | 74.58 |
| | FedMeta | 82.69 | 48.44 | 68.84 | 71.82 | 74.14 |
| | q-FFL | 82.14 | 73.10 | 68.22 | 70.64 | 73.77 |
| | CoPreFL-SGD ($\gamma = 0.25$) | 83.63 | 41.73 | 69.76 | 73.46 | 75.64 |
| | CoPreFL ($\gamma = 0.25$) | **86.63** | **31.58** | **73.05** | **75.82** | **77.58** |

Table 3: **Scenario II (CIFAR-100):** Average performance across 10 non-IID downstream FL tasks, initialized with various FL pre-trained models.

**(a) Dataset: CIFAR-100; Scenario: I**  **(b) Dataset: CIFAR-100; Scenario: II**  **(c) Dataset: Tiny-ImageNet; Scenario: I**  **(d) Dataset: Tiny-ImageNet; Scenario: II**

Figure 2: The distributions of testing accuracy in non-IID FL downstream tasks under various non-IID pre-training scenarios. `CoPreFL` achieves the best average accuracy and a more centralized (i.e., fairer) distribution while also improving the accuracies of worst performing clients.

**Effect of balancer $\gamma$ in `CoPreFL`.** Table 4 displays the average performance across 10 non-IID FL downstream tasks initialized with our `CoPreFL` method using different balancers $\gamma$ on Tiny-ImageNet. A larger $\gamma$ implies that the pre-trained model prioritizes the devices' average performance, whereas a smaller $\gamma$ implies that the pre-trained model aims to promote greater uniformity in the training accuracy distribution.

| $\gamma$ of `CoPreFL` | Acc ↑ | Variance ↓ |
|---|---|---|
| 0.0 | 83.11 | 24.70 |
| 0.25 | 84.04 | 35.88 |
| 0.5 | 85.23 | 35.40 |
| 0.75 | 85.19 | 39.31 |
| 1.0 | 86.33 | 39.81 |

Table 4: Effect of $\gamma$ in non-IID scenario I, using Tiny-ImageNet.

In Table 4, as the balancer of the pre-trained model increases, we observe an increase in the average accuracy of downstream FL tasks but a decrease in fairness, indicated by higher variance. The trend shows that we successfully establish a robust pre-training environment that mimics downstream FL tasks and addresses the challenges related to unseen scenarios and unfair predictions. The design and control of objective functions (3) and (4) during the pre-training phase can yield implicit benefits for achieving goals (1) and (2) when utilizing these pre-trained models as an initialization for downstream FL tasks.

**Other initialization methods.** In addition to FL algorithms, in Table 5, we consider other initialization methods with random weights or a centralized pre-trained model. In scenario I, we collect a centralized dataset from all the clients in the pre-training phase, while in scenario II, we gather the dataset from both the clients and the server for centralized pre-training. While the centralized method improves the accuracy of downstream FL compared to random initialization, it introduces

| Pre-training | | Downstream: non-IID FL | |
|---|---|---|---|
| **Scenario** | **Method** | **Acc ↑** | **Variance ↓** |
| | Random | 75.32 | 41.39 |
| I | Centralized | 81.30 | 69.44 |
| | CoPreFL | **83.29** | **34.69** |
| | Random | 77.50 | 53.00 |
| II | Centralized | 82.07 | 70.90 |
| | CoPreFL | **86.63** | **31.58** |

Table 5: Comparison with other initialization methods on CIFAR-100.

significant variance across clients, resulting in fairness issues. Our method outperforms the centralized baseline, demonstrating that proper pre-training designs using FL, aligned with controlled objectives (3) and (4), can indeed enhance FL itself through initialization. Additional results for these initialization methods are provided in Appendix C.4 under different settings, consistently confirming the advantage of `CoPreFL`. In addition to the baselines introduced in Section 4, we also include FedDyn (Acar et al., 2021), designed to address non-IID issues, for a detailed comparison. Moreover, we also explore a scenario that involves pre-training with large public datasets. The detail and the results can also be found in Appendix C.4.

**Additional experimental results.** A scenario where the downstream task contains some seen classes from the pre-training phase is discussed in Appendix C.6. Moreover, the results for different downstream FL algorithms are presented in Appendix C.7. These additional results show the superiority of our method.

## 5 CONCLUSION

We introduced `CoPreFL`, a collaborative pre-training method aimed at providing any unseen FL task with a robust initialization. Our pre-training approach takes into account practical scenarios where data may be distributed across the clients and the server during pre-training. `CoPreFL` leverages meta-learning to empower the pre-trained model with the ability to handle unseen data while striking a balance between average performance and fairness. Extensive experiments in various setups demonstrate that our pre-training method serves as a dependable initialization for any FL task, enabling them to achieve superior average performance and more fair predictions across the clients.

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

## A    DETAILED PROCEDURE FOR OUR CoPreFL IN SCENARIO II

This section provides a detailed introduction to our CoPreFL in scenario II, as discussed in Section 3.3. Similar to the goals of CoPreFL in scenario II, we still aim to balance between objective functions (3) and (4), but in this scenario, the data used to perform meta-updates and control our objectives is different. During each federated round $t$ in the pre-training phase, participants download the global model $\Phi^{t-1}$ from the previous round (line 6 in Algorithm 2) . Subsequently, they perform few local training iterations utilizing their respective local datasets $D_j^p$ (line 7 in Algorithm 2). This process leads to a training loss $\mathcal{L}_{D_j^p}(\Phi^t)$, defined as $\frac{1}{|D_j^p|} \sum_{(x,y) \in D_j^p} \ell(\Phi^{t-1}(x), y)$, where $x$ represents the input (e.g., images), $y$ denotes the true label, and $\ell(\cdot)$ denotes the loss function (e.g.,cross-entropy loss). The local models are then updated based on this loss, yielding their respective updated local models $\Phi_j^t$ (line 8 in Algorithm 2). Upon the completion of local training by all participants, their models are transmitted to the server (line 10 in Algorithm 2), and the server aggregates these models into a temporary global model $\overline{\Phi^t} = \sum_{j \in m} \mu_j \Phi_j^t$, which is weighted by relative dataset sizes $\mu_j = \frac{|D_j^p|}{\sum_{i \in m} |D_i^p|}$.

We then perform meta-updates for the temporary global model $\overline{\Phi^t}$ using server's dataset $D^s$. To start, we first randomly divide the server's dataset $D^s$ into $|m|$ equal partitions. Subsequently, the server evaluates the temporary global model $\overline{\Phi^t}$ using each subset $D_j^s$ (line 12 in Algorithm 2), resulting in the corresponding meta-loss $\mathcal{L}_{D_j^s}(\overline{\Phi^t}) = \frac{1}{|D_j^s|} \sum_{(x,y) \in D_j^s} \ell(\overline{\Phi^t}(x), y)$. The collective server's meta-loss, denoted as $\mathcal{L}_{D^s}(\overline{\Phi^t})$ in line 14 of Algorithm 2, is determined by aggregating all the meta-loss values obtained from $D_j^s$, and we also calculate the variance $\sigma_{D^s}^2 = \frac{1}{m} \sum_{i \in m}(\mathcal{L}_{D_i^s}(\overline{\Phi^t}) - \frac{1}{m}\mathcal{L}_{D^s}(\overline{\Phi^t}))$ across server's meta-losses to examine the performance distribution. We then tailor a customized server meta-loss $\mathcal{L}_{meta}(\overline{\Phi^t}) = \gamma \mathcal{L}_{D^s}(\overline{\Phi^t}) + (1 - \gamma)\sigma_{D^s}^2(\overline{\Phi^t})$ to achieve a balance between optimizing for performance and fairness. Finally, in line 17 of Algorithm 2, we employ the customized server meta-loss $\mathcal{L}_{meta}(\overline{\Phi^t})$ to update the temporary global model $\overline{\Phi^t}$, aligning it with our controlled objective. The server then sends this meta-updated global model $\Phi^t$ to the participants in the next round for initialization. After completing $T$ federated rounds, we regard the final global model $\Phi^T$ as the pre-trained model in scenario II, which serves as the initialization for the fine-tuning phase of the diverse FL tasks.

---

**Algorithm 2** Our Pre-training Method CoPreFL (Pre-training Phase in Scenario II)

1: **Input:** $M$ clients in the pre-training phase, with each client $i$ holding their own dataset $D_i^p$; the server also holds a dataset $D^s$.
2: **for** Each communication round $t = 1, 2, ..., T$ **do**
3:     Randomly select a set of client $m \subset M$ to participate in learning
4:     Randomly split server's dataset $D^s$ into $|m|$ subsets
5:     **for** Each participant $j$ in parallel **do**
6:         Downloads $\Phi^{t-1}$ from the server
7:         $\mathcal{L}_{D_j^p}(\Phi^t) \leftarrow \frac{1}{|D_j^p|} \sum_{(x,y) \in D_j^p} \ell(\Phi^{t-1}(x), y)$     ▷ Get local loss using local dataset $D_j^p$
8:         $\Phi_j^t \leftarrow \Phi^{t-1} - \eta\nabla\mathcal{L}_{D_j^p}(\Phi^t)$     ▷ Perform SGD local update using local loss $\mathcal{L}_{D_j^p}$
9:     **end for**
10:     $\overline{\Phi^t} \leftarrow \sum_{j \in m} \frac{|D_j^p|}{\sum_{i \in m} |D_i^p|} \Phi_j^t$     ▷ Model aggregation to construct temporary global model
11:     **for** Each split server's dataset $D_j^s$ in parallel, Server **do**
12:         $\mathcal{L}_{D_j^s}(\overline{\Phi^t}) \leftarrow \frac{1}{|D_j^s|} \sum_{(x,y) \in D_j^s} \ell(\overline{\Phi^t}(x), y)$     ▷ Server's meta-loss corresponding to $D^s$
13:     **end for**
14:     Overall meta-loss on server: $\mathcal{L}_{D^s}(\overline{\Phi^t}) = \sum_{j \in m} \mathcal{L}_{D_j^s}(\overline{\Phi^t})$
15:     Variance across server meta-losses: $\sigma_{D^s}^2(\overline{\Phi^t}) = \frac{1}{|m|} \sum_{j \in m}(\mathcal{L}_{D_j^s}(\overline{\Phi^t}) - \frac{1}{|m|}\mathcal{L}_{D^s}(\overline{\Phi^t}))^2$
16:     Customized server meta-loss: $\mathcal{L}_{meta}(\overline{\Phi^t}) = \gamma\mathcal{L}_{D^s}(\overline{\Phi^t}) + (1 - \gamma)\sigma_{D^s}^2(\overline{\Phi^t})$
17:     $\Phi^t \leftarrow \overline{\Phi^t} - \zeta\nabla\mathcal{L}_{meta}(\overline{\Phi^t})$     ▷ Model updates using customized loss
18: **end for**
19: **Output:** A pre-trained model for downstream FL tasks: $\Phi^T$

---

# B  DETAILED SETTINGS FOR DATASETS, HYPERPARAMETERS, AND DOWNSTREAM TASK

## B.1  DATASETS DETAILS

For CIFAR-100, the dataset is divided into 80 classes for pre-training and 20 classes for downstream FL task, while for Tiny-ImageNet, the dataset is separated into 160 classes for pre-training and 40 classes for downstream FL task. This is to model a practical scenario where labels at downstream tasks are not known and not available during pre-training. We randomly select 95% of the samples from the pre-training dataset to form the dataset for clients, while the remaining 5% of samples constitute the server dataset. For FEMNIST, we report the detailed settings and results in Appendix C.5.

## B.2  HYPERPARAMETERS AND COMPUTE SETTINGS

For our method, the SGD optimizer with a learning rate of $\eta = 10^{-3}$ and $\zeta = 10^{-3}$ is adopted for both local and meta updates. Both local and meta learning rates are searched within the range of [1e-2, 5e-3, 1e-3, 5e-4]. We searched for learning rates within the range of [1e-2, 5e-3, 1e-3, 5e-4] for local training of all FL pre-training baselines and selected 1e-3 as the optimal learning rate for them. In scenario II, each FL baseline will continue to conduct a few SGD iterations using the server's data after constructing their global model. We searched for learning rates in the range of [1e-2 and 1e-3] for this additional training and selected 1e-3 as the optimal learning rate for the server. Regarding hyperparameters in the q-FFL baseline, we conducted experiments with q-values of 1, 3, and 5 and reported the corresponding best statistics. We select a learning rate $\eta$ from the range [1e-2, 5e-3, 1e-3, 5e-4] for local updates in our `CoPreFL` and determined that 1e-3 provides the best results. Additionally, for meta-updates in both scenarios, we search for the learning rate $\zeta$ within the range [1e-2, 1e-3] and find that 1e-3 is the optimal value. In the case of the centralized baseline mentioned in Section 4.2, we searched for the optimal learning rate within the range [1e-2, 5e-3, 1e-3, 5e-4, 1e-4], ultimately selecting 1e-3. We utilized the SGD optimizer for all updates across all methods, and the batch size is set to be 32 for all experiments. In our simulations of `CoPreFL`, we assessed various balancer values $\gamma$ from the range [0.0, 0.25, 0.5, 0.75, 1.0] in all scenarios and reported the best-performing value in our paper. We run all experiments on a 3-GPU cluster of Tesla V100 GPUs, with each GPU having 32GB of memory.

## B.3  DETAILS FOR DOWNSTREAM TASK

To generate each downstream FL task, we randomly select 5 classes out of the 20 classes from the CIFAR-100 dataset and 40 classes from the Tiny-ImageNet dataset, and distribute the corresponding data samples to $|G| = 10$ clients following either IID or non-IID data distributions. It is important to note that these classes (20 and 40 classes) are distinct from those used in the pre-training phase. Each participant in the downstream phase utilizes 80% of its local data as training samples, while the remaining 20% is reserved for testing samples. To see the impact of different pre-training methods, we fix the training procedure at each downstream task to the commonly adopted FedAvg algorithm. We consider $R = 50$ FL rounds using the training set, involving 5 iterations per round for local training using the SGD optimizer with a learning rate of $10^{-3}$. We evaluate the final global model's performance using testing samples from each participant and report both the accuracy and the variance of the accuracy distribution across $|G|$ clients for each FL task. We consider a total of $X = 10$ downstream FL tasks, and the evaluation metrics are reported as the average across these $X$ downstream FL tasks.

# C  ADDITIONAL EXPERIMENTS AND ANALYSES

## C.1  DOWNSTREAM FL RESULTS WITH SCENARIO I PRE-TRAINING

This section provides supplementary results for pre-training scenario I, as discussed in Section 4.2. We train pre-trained models using both IID and non-IID distributions, varying the number of participants in each federated round during the pre-training phase. To be more specific, we specify the number of participants $|m|$ as 15, 20, 25, and 30 out of 100 clients to participate in FL during the pre-training phase. Subsequently, we evaluate these pre-trained models by initializing them for IID

and non-IID downstream FL tasks. Tables 6, 7, 8, and 9 display the average performance across 10 **IID** FL downstream tasks and Tables 10, 11, 12, and 13 show the average performance across 10 **non-IID** FL downstream tasks. In both cases, the downstream FL were initialized by pre-trained models trained on 15, 20, 25, and 30 participants out of 100 clients, respectively, on the CIFAR-100 dataset. For the Tiny-ImageNet dataset, Tables 14, 15, 16, and 17 show the average performance across 10 **IID** FL downstream tasks and Tables 18, 19, 20, and 21 display the average performance across 10 **non-IID** FL downstream tasks. In both cases, the downstream FL were also initialized by pre-trained models trained on 15, 20, 25, and 30 participants out of 100 clients, respectively. Across these experimental results, considering different data distribution setup during the pre-training phase and different datasets, our `CoPreFL` consistently demonstrates superiority over the baseline when used as an initialization for various downstream FL tasks. By creating an environment that mimics downstream FL tasks and specifically addressing the challenges encountered in these tasks, our designed pre-training objectives (3) and (4) establish an ideal pre-trained model for FL. As initialization for various unseen FL tasks, our `CoPreFL` provide downstream FL tasks with both better average performance and fairer predictions across clients.

## C.2 DOWNSTREAM FL RESULTS WITH SCENARIO II PRE-TRAINING

This section provides supplementary results for scenario II, where the server holds a small portion of the dataset during the pre-training phase. We also consider varying numbers of participants $|m|$, specifically 15, 20, 25, and 30 out of 100 clients, during the pre-training phase for these models. Tables 22, 23, 24, and 25 display the average performance across 10 **IID** FL downstream tasks and Tables 26, 27, 28, and 29 show the average performance across 10 **non-IID** FL downstream tasks. In both cases, the downstream FL were initialized by pre-trained models trained on 15, 20, 25, and 30 participants out of 100 clients, respectively, on the CIFAR-100 dataset. For the Tiny-ImageNet dataset, Tables 30, 31, 32, and 33 show the average performance across 10 **IID** FL downstream tasks and Tables 34, 35, 36, and 37 display the average performance across 10 **non-IID** FL downstream tasks. In both cases, the downstream FL were also initialized by pre-trained models trained on 15, 20, 25, and 30 participants out of 100 clients, respectively. It's important to note that in this scenario, FedAvg, FedMeta, and q-FFL undergo further training using server data through the SGD optimizer after each method completes its local iterations and obtains its respective global model in each round (Yang et al., 2023; Bian et al., 2023). Similarly, `CoPreFL-SGD` is trained using server data with the SGD optimizer on $\Phi^t$ in line 17 of Algorithm 1, while `CoPreFL` follows Algorithm 2, utilizing server data for meta-updates. By incorporating meta-updates using server data to align with our objectives (3) and (4), our pre-training method consistently outperforms other baselines, leading to improved average accuracy and reduced variance. Comparing `CoPreFL` with `CoPreFL-SGD` strongly suggests that, rather than conducting a few SGD iterations using server data, which may dilute our objectives, we recommend building pre-training objectives upon server data using meta-updates.

## C.3 TESTING ACCURACY DISTRIBUTION OF DOWNSTREAM FL TASKS

This section presents supplementary distribution results to evaluate the fairness of the pre-trained models discussed in Section 4.2. For pre-trained models trained in scenario I, Figures 3 and 4 show the testing accuracy distribution of IID and non-IID FL tasks on CIFAR-100 dataset, and Figures 5 and 6 display the respective distribution on Tiny-ImageNet dataset. Figures 7 and 8 present the testing accuracy distribution of IID and non-IID FL tasks initialized by pre-trained models trained in scenario II on CIFAR-100 dataset, and Figures 9 and 10 show the respective distribution on Tiny-ImageNet dataset. Across our experimental results, which encompass different data distribution setups and scenarios during the pre-training phase and various datasets, our `CoPreFL` consistently enhances the fairness of testing accuracy distributions for diverse downstream FL tasks. In general, distributions of FL tasks initialized by our `CoPreFL` tend to shift towards the right, indicating improved prediction performance. Moreover, when analyzing clients positioned at the left end of the distribution in each pre-training method, our approach effectively elevates underperforming clients towards the right end, resulting in enhanced predictive accuracy for these clients.

## C.4 DIFFERENT INITIALIZATION METHODS

This section presents supplementary results with different initialization methods discussed in Section 4.2, including random initialization and centralized model initialization for downstream FL. Table 38 shows the performance of IID FL downstream tasks initialized by different pre-training methods trained in two scenarios on CIFAR-100 dataset. Tables 39 and 40 present the performance of IID and non-IID FL downstream tasks initialized by different pre-training methods trained on Tiny-ImageNet dataset. We select $|m| = 20$ for CoPreFL in Tables 38, 39, and 40. Comparing centralized and random initialization, we observe that the centralized method generally improves the average accuracy of downstream FL but at the cost of higher variance in most cases. However, our CoPreFL consistently enhances both average accuracy and fairness in various downstream FL tasks, demonstrating that with proper FL designs as pre-trained model, FL can be improved through initialization.

In addition to utilizing CIFAR-100 and Tiny-ImageNet datasets, where we partition our data for pre-training and downstream tasks, we also explore a scenario where public large datasets are available for pre-training phase. We conducted experiments using pre-trained models with the ImageNet dataset (Deng et al., 2009), a widely used large public dataset for image classification. We sampled 200 images for each of the 1000 classes in ImageNet_1K. Both a centralized model and our proposed CoPreFL were pre-trained using ImageNet_1K, and we initialized downstream FL tasks using these methods. For the pre-training phase, we implemented a centralized model with the SGD optimizer and a learning rate of 1e-3, training the model for 50 epochs. In our proposed method, we distributed all the data across $|M| = 100$ clients, sampling $|m| = 20$ clients in each round, and conducted CoPreFL for 50 rounds. Table 41 shows the performance of non-IID FL downstream tasks on CIFAR-100 dataset, where the downstream tasks are initialized by different methods trained on ImageNet dataset. It's important to note that there must be overlapping classes between ImageNet_1K and the 20-class downstream dataset in CIFAR-100. The results indicate that employing a large public dataset for pre-training can enhance overall accuracy performance. However, the centralized method still introduces larger performance variance in downstream FL compared to our method.

Additionally, we establish a non-IID-related baseline for the pre-training phase for a detailed comparison. Table 42 presents the comparison between FedDyn (with an $\alpha$ parameter set to 0.01) and our CoPreFL. We can see that, despite its consideration of non-IIDness in design, FedDyn cannot provide superiority over our method in downstream FL tasks due to the absence of considerations for unseen adaptation and performance fairness during the pre-training phase.

Furthermore, we introduce a personalized FL baseline, Per-FedAvg (Fallah et al., 2020), during the pre-training phase to offer a comparison with another FL approach that incorporates meta-learning. We follow a two-step gradient descent for local client training introduced in Per-FedAvg, while utilizing the final global model as the initialization for our downstream FL task. Table 49 presents a comparison between our method and Per-FedAvg, with the results demonstrating the superiority of our method.

## C.5 ADDITIONAL RESULTS FOR FEMNIST DATASET

We also consider the FEMNIST dataset, widely used in FL research, following the data partition provided in (Park et al., 2021). We divide the 62 classes into 52 alphabet classes for the pre-training phase, reserving the remaining 10 digit classes for downstream FL tasks. Instead of using a ResNet-18 model, we employ a model consisting of two $3\times3$ convolutional layers followed by two linear layers. We fixed the total number of clients as $|M| = 100$ for pre-training and $|G| = 10$ for downstream FL tasks. During the pre-training phase, we set the number of participants $|m| = 20$ and the federated round $T = 50$ for each pre-trained method. We use the same learning rate and optimizer introduced in Appendix B.2. For downstream tasks, we randomly select 5 classes from 10 classes for downstream task to conduct each FL task using FedAvg, and we perform $X = 10$ FL tasks with federated round $R = 10$. Tables 43 and 44 display the averaged performance of 10 IID and 10 non-IID FL downstream tasks, initialized by various pre-training methods trained in scenario I, on the FEMNIST dataset. For scenario II, Tables 45 and 46 show the performance of IID and non-IID downstream FL tasks. The results also demonstrate that our proposed CoPreFL serves as a robust initialization for various FL setups, benefiting both performance and fairness.

## C.6 RESULTS FOR BOTH UNSEEN/SEEN CLASSES IN DOWNSTREAM FL TASKS

In Table 47, we consider a scenario where the downstream task contains some overlapped classes with the pre-training phase. Specifically, we continue to use the data from 80 classes in the CIFAR-100 dataset for pre-training. Then, we sample 10 classes from this 80-class pre-training dataset and 10 classes from the original 20-class downstream dataset for this scenario. This results in a new downstream dataset with 50% of seen data. We randomly select 5 classes from this new downstream dataset for each FL task, repeating the process 10 times and averaging the performance. The results demonstrate that each pre-trained model performs better in this seen/unseen downstream scenario compared to the results in Table 1. The observed trend aligns consistently with the findings in Tables 1, confirming the advantage of the proposed method.

## C.7 RESULTS FOR DIFFERENT DOWNSTREAM FL TASK

In addition to the general downstream FL tasks built by FedAvg, we consider FedProx, a more advanced FL algorithm that addresses heterogeneity compared to FedAvg, to examine the robustness and generalizability of our pre-trained method. In Table 48, the results demonstrate that our pre-trained method maintains superiority in downstream FedProx compared to other pre-training methods. It's important to note that the choice of FedAvg as our downstream task is made to minimize the varying impact introduced by other FL algorithms. Comparing the pre-training + downstream pairs, the improvement of CorPreFL + FedAvg (in Table 1) over Centralized + FedProx (in Table 48) shows that a better initialization, which considers the distributed scenario and balances fairness/performance in the pre-training phase, could potentially benefit the inferior downstream FL algorithm.

## C.8 KEY APPLICATIONS

Consider a healthcare application where each client, such as a hospital or an individual patient, aims to build a comprehensive global model capable of classifying a wide range of diseases. However, individual clients may possess limited types of diseases in their local datasets – for instance, one client may have data on diseases A and B but lacks information on diseases C and D. In this context, federated learning becomes essential. Clients need to collaborate to construct a global model that not only reflects the diseases available locally but also incorporates information about diseases not present in their individual datasets, ensuring a more robust and universally applicable healthcare model. Similarly, in the domain of autonomous vehicles, each self-driving car may strive to develop a global model for scenario detection in various weather conditions. However, individual cars might encounter limited weather scenarios locally – one car might navigate through a desert environment, while another faces challenges in a snowy storm. Through federated learning, these cars can collectively construct a global model that accounts for a broad spectrum of weather conditions, ensuring robust scenario detection capabilities for all vehicles involved.

As noted in Remark 2, the server can intentionally partition the centralized dataset and implement our scheme, utilizing multiple computing units available at the server, to obtain a pre-trained model. The advantage of this approach, compared to simple centralized training, lies in mitigating side effects such as performance biases and the substantial variance associated with centralized training. This phenomenon stems from the lack of generalizability in the model's design. When a model undergoes pre-training in a centralized manner based on SGD, it becomes rigidly bound to the knowledge in the pre-training dataset. This fixation presents a challenge in adapting the model to the diverse clients that may possess new or unseen data in downstream tasks. Such variations can arise from factors like the time-varying environment or new clients joining the system, as exemplified in the aforementioned applications: classifying different scenarios based on the self-driving car's environment, identifying diverse diseases based on patient interests, or enabling face/speech recognition for new phone users.

| Pre-training (Scenario I, $|m| = 15$) | | Downstream: IID FL | | | | |
|---|---|---|---|---|---|---|
| **Distribution** | **Method** | **Acc ↑** | **Variance ↓** | **Worst 10% ↑** | **Worst 20% ↑** | **Worst 30% ↑** |
| IID | FedAvg | 87.01 | 15.44 | 78.91 | 80.55 | 81.41 |
| | FedMeta | 87.09 | 14.67 | 81.45 | 82.42 | 83.15 |
| | q-FFL | 87.25 | 13.25 | 80.85 | 81.52 | 82.26 |
| | CoPreFL ($\gamma = 0.5$) | **87.84** | **11.49** | **82.61** | **83.52** | **84.44** |
| Non-IID | FedAvg | 85.85 | 15.37 | 78.91 | 80.55 | 81.41 |
| | FedMeta | 86.84 | 12.25 | 81.45 | 82.42 | 83.15 |
| | q-FFL | 86.37 | 13.54 | 80.85 | 81.52 | 82.26 |
| | CoPreFL ($\gamma = 0.25$) | **86.90** | **8.70** | **81.52** | **82.58** | **83.21** |

Table 6: Average performance across 10 **IID** downstream FL tasks, initialized with various FL pre-trained methods using **15** out of 100 participants in **scenario I**, on the **CIFAR-100** dataset.

| Pre-training (Scenario I, $|m| = 20$) | | Downstream: IID FL | | | | |
|---|---|---|---|---|---|---|
| **Distribution** | **Method** | **Acc ↑** | **Variance ↓** | **Worst 10% ↑** | **Worst 20% ↑** | **Worst 30% ↑** |
| IID | FedAvg | 87.34 | 12.46 | 80.48 | 81.64 | 82.51 |
| | FedMeta | 86.70 | 14.52 | 81.33 | 82.06 | 82.75 |
| | q-FFL | 86.95 | 11.97 | 80.48 | 81.58 | 82.51 |
| | CoPreFL ($\gamma = 0.5$) | **87.54** | **10.18** | **81.94** | **82.97** | **83.84** |
| Non-IID | FedAvg | 86.04 | 14.36 | 80.85 | 81.39 | 82.02 |
| | FedMeta | 86.15 | 16.16 | 79.52 | 81.27 | 82.18 |
| | q-FFL | 86.30 | 17.14 | 80.24 | 81.58 | 82.46 |
| | CoPreFL ($\gamma = 0.25$) | **86.32** | **14.14** | **81.45** | **82.20** | **82.75** |

Table 7: Average performance across 10 **IID** downstream FL tasks, initialized with various FL pre-trained methods using **20** out of 100 participants in **scenario I**, on the **CIFAR-100** dataset.

| Pre-training (Scenario I, $|m| = 25$) | | Downstream: IID FL | | | | |
|---|---|---|---|---|---|---|
| **Distribution** | **Method** | **Acc ↑** | **Variance ↓** | **Worst 10% ↑** | **Worst 20% ↑** | **Worst 30% ↑** |
| IID | FedAvg | 87.68 | 11.36 | 81.58 | 82.79 | 83.68 |
| | FedMeta | 87.10 | 13.62 | 81.45 | 82.61 | 83.23 |
| | q-FFL | 87.07 | 17.89 | 80.48 | 81.82 | 82.59 |
| | CoPreFL ($\gamma = 0.0$) | **88.13** | **9.30** | **82.85** | **83.94** | **84.75** |
| Non-IID | FedAvg | 86.78 | 11.90 | 81.09 | 81.82 | 82.55 |
| | FedMeta | 85.41 | 15.05 | 79.15 | 79.94 | 80.81 |
| | q-FFL | 85.92 | 12.11 | 79.03 | 80.55 | 81.49 |
| | CoPreFL ($\gamma = 0.0$) | **86.84** | **11.16** | **82.06** | **82.85** | **83.43** |

Table 8: Average performance across 10 **IID** downstream FL tasks, initialized with various FL pre-trained methods using **25** out of 100 participants in **scenario I**, on the **CIFAR-100** dataset.

| Pre-training (Scenario I, $|m| = 30$) | | Downstream: IID FL | | | | |
|---|---|---|---|---|---|---|
| **Distribution** | **Method** | **Acc ↑** | **Variance ↓** | **Worst 10% ↑** | **Worst 20% ↑** | **Worst 30% ↑** |
| IID | FedAvg | 86.52 | 13.54 | 80.85 | 81.76 | 82.55 |
| | FedMeta | 87.65 | 13.47 | 81.58 | 82.91 | 83.64 |
| | q-FFL | 86.40 | 15.68 | 79.27 | 80.12 | 81.45 |
| | CoPreFL ($\gamma = 0.0$) | **87.90** | **11.16** | **82.67** | **83.58** | **84.36** |
| Non-IID | FedAvg | 86.78 | 12.32 | 80.85 | 81.76 | 82.55 |
| | FedMeta | 85.87 | 17.22 | 81.58 | 82.09 | 82.64 |
| | q-FFL | 85.77 | 13.40 | 79.27 | 80.12 | 81.45 |
| | CoPreFL ($\gamma = 0.5$) | **87.04** | **9.18** | **81.70** | **82.12** | **82.91** |

Table 9: Average performance across 10 **IID** downstream FL tasks, initialized with various FL pre-trained methods using **30** out of 100 participants in **scenario I**, on the **CIFAR-100** dataset.

| Pre-training (Scenario I, $|m| = 15$) | | Downstream: Non-IID FL | | | | |
|---|---|---|---|---|---|---|
| **Distribution** | **Method** | **Acc ↑** | **Variance ↓** | **Worst 10% ↑** | **Worst 20% ↑** | **Worst 30% ↑** |
| IID | FedAvg | 81.46 | 62.09 | 68.87 | 71.12 | 72.76 |
| | FedMeta | 81.20 | 63.84 | 69.39 | 71.52 | 73.14 |
| | q-FFL | 83.45 | 39.94 | 69.95 | 73.66 | 75.43 |
| | CoPreFL ($\gamma = 0.75$) | **84.79** | **37.09** | **72.71** | **74.80** | **76.75** |
| Non-IID | FedAvg | 83.76 | 51.84 | 69.50 | 72.80 | 74.30 |
| | FedMeta | 82.65 | 39.19 | 69.39 | 72.76 | 74.87 |
| | q-FFL | 82.00 | 53.00 | 70.78 | 73.03 | 74.22 |
| | CoPreFL ($\gamma = 0.25$) | **84.55** | **38.07** | **71.47** | **73.40** | **75.20** |

Table 10: Average performance across 10 **non-IID** downstream FL tasks, initialized with various FL pre-trained methods using **15** out of 100 participants in **scenario I**, on the **CIFAR-100** dataset.

| Pre-training (Scenario I, $|m| = 20$) | | Downstream: Non-IID FL | | | | |
|---|---|---|---|---|---|---|
| **Distribution** | **Method** | **Acc ↑** | **Variance ↓** | **Worst 10% ↑** | **Worst 20% ↑** | **Worst 30% ↑** |
| IID | FedAvg | 84.20 | 57.15 | 68.43 | 71.83 | 74.38 |
| | FedMeta | 83.80 | 42.64 | 72.30 | 73.79 | 75.47 |
| | q-FFL | 82.60 | 45.56 | 70.46 | 73.41 | 75.14 |
| | CoPreFL ($\gamma = 0.25$) | **84.36** | **38.56** | **73.66** | **75.63** | **77.40** |
| Non-IID | FedAvg | 78.96 | 64.80 | 62.70 | 67.00 | 69.80 |
| | FedMeta | 82.45 | 48.72 | 68.97 | 72.41 | 74.35 |
| | q-FFL | 80.01 | 88.92 | 64.39 | 67.48 | 70.30 |
| | CoPreFL ($\gamma = 0.75$) | **83.29** | **34.69** | **71.58** | **73.20** | **74.59** |

Table 11: Average performance across 10 **non-IID** downstream FL tasks, initialized with various FL pre-trained methods using **20** out of 100 participants in **scenario I**, on the **CIFAR-100** dataset.

| Pre-training (Scenario I, $|m| = 25$) | | Downstream: Non-IID FL | | | | |
|---|---|---|---|---|---|---|
| **Distribution** | **Method** | **Acc ↑** | **Variance ↓** | **Worst 10% ↑** | **Worst 20% ↑** | **Worst 30% ↑** |
| IID | FedAvg | 84.02 | 51.98 | 71.26 | 73.21 | 75.57 |
| | FedMeta | 82.44 | 55.06 | 68.53 | 71.73 | 73.95 |
| | q-FFL | 82.63 | 47.20 | 70.52 | 72.42 | 74.01 |
| | CoPreFL ($\gamma = 0.75$) | **85.60** | **37.45** | **74.42** | **76.53** | **78.43** |
| Non-IID | FedAvg | 82.01 | 39.82 | 70.75 | 73.02 | 74.63 |
| | FedMeta | 84.02 | 39.56 | 71.86 | 75.17 | 76.80 |
| | q-FFL | 82.18 | 46.79 | 70.53 | 72.43 | 73.61 |
| | CoPreFL ($\gamma = 0.25$) | **85.72** | **29.38** | **75.81** | **77.24** | **78.54** |

Table 12: Average performance across 10 **non-IID** downstream FL tasks, initialized with various FL pre-trained methods using **25** out of 100 participants in **scenario I**, on the **CIFAR-100** dataset.

| Pre-training (Scenario I, $|m| = 30$) | | Downstream: Non-IID FL | | | | |
|---|---|---|---|---|---|---|
| **Distribution** | **Method** | **Acc ↑** | **Variance ↓** | **Worst 10% ↑** | **Worst 20% ↑** | **Worst 30% ↑** |
| IID | FedAvg | 79.60 | 83.36 | 62.68 | 65.87 | 68.57 |
| | FedMeta | 79.90 | 48.02 | 67.01 | 69.69 | 71.46 |
| | q-FFL | 83.02 | 52.27 | 70.64 | 72.71 | 74.50 |
| | CoPreFL ($\gamma = 0.75$) | **83.48** | **45.16** | **70.80** | **72.72** | **74.59** |
| Non-IID | FedAvg | 81.79 | 50.84 | 69.70 | 72.08 | 74.20 |
| | FedMeta | 82.61 | 43.43 | 71.84 | 73.30 | 74.68 |
| | q-FFL | 82.68 | 54.17 | 68.68 | 72.08 | 74.06 |
| | CoPreFL ($\gamma = 0.75$) | **83.48** | **40.20** | **72.83** | **74.29** | **75.80** |

Table 13: Average performance across 10 **non-IID** downstream FL tasks, initialized with various FL pre-trained methods using **30** out of 100 participants in **scenario I**, on the **CIFAR-100** dataset.

| Pre-training (Scenario I, $|m| = 15$) | | Downstream: IID FL | | | | |
|---|---|---|---|---|---|---|
| Distribution | Method | Acc ↑ | Variance ↓ | Worst 10% ↑ | Worst 20% ↑ | Worst 30% ↑ |
| IID | FedAvg | 84.27 | 16.48 | 77.20 | 78.43 | 79.27 |
| | FedMeta | 84.15 | 19.27 | 78.50 | 79.65 | 80.66 |
| | q-FFL | 84.24 | 17.64 | 77.20 | 78.86 | 79.99 |
| | CoPreFL ($\gamma = 0.0$) | **85.05** | **15.21** | **79.08** | **80.38** | **81.19** |
| Non-IID | FedAvg | 85.19 | 15.13 | 77.78 | 79.29 | 80.38 |
| | FedMeta | 85.35 | 15.60 | 78.50 | 80.01 | 81.00 |
| | q-FFL | 85.91 | 15.76 | 78.22 | 80.38 | 81.12 |
| | CoPreFL ($\gamma = 0.0$) | **86.39** | **10.63** | **79.08** | **80.45** | **81.24** |

Table 14: Average performance across 10 **IID** downstream FL tasks, initialized with various FL pre-trained methods using **15** out of 100 participants in **scenario I**, on the **Tiny-ImageNet** dataset.

| Pre-training (Scenario I, $|m| = 20$) | | Downstream: IID FL | | | | |
|---|---|---|---|---|---|---|
| Distribution | Method | Acc ↑ | Variance ↓ | Worst 10% ↑ | Worst 20% ↑ | Worst 30% ↑ |
| IID | FedAvg | 85.74 | 17.39 | 77.20 | 78.79 | 79.80 |
| | FedMeta | 85.56 | 17.64 | 77.92 | 79.51 | 80.57 |
| | q-FFL | 84.64 | 21.07 | 78.79 | 79.74 | 80.91 |
| | CoPreFL ($\gamma = 0.5$) | **86.03** | **13.99** | **79.08** | **80.09** | **81.24** |
| Non-IID | FedAvg | 85.43 | 17.31 | 77.49 | 79.65 | 80.76 |
| | FedMeta | 84.16 | 16.89 | 77.20 | 79.73 | 81.05 |
| | q-FFL | 85.83 | 19.18 | 78.07 | 79.37 | 80.62 |
| | CoPreFL ($\gamma = 0.5$) | **86.00** | **16.16** | **79.37** | **80.30** | **81.19** |

Table 15: Average performance across 10 **IID** downstream FL tasks, initialized with various FL pre-trained methods using **20** out of 100 participants in **scenario I**, on the **Tiny-ImageNet** dataset.

| Pre-training (Scenario I, $|m| = 25$) | | Downstream: IID FL | | | | |
|---|---|---|---|---|---|---|
| Distribution | Method | Acc ↑ | Variance ↓ | Worst 10% ↑ | Worst 20% ↑ | Worst 30% ↑ |
| IID | FedAvg | 85.24 | 19.71 | 78.35 | 79.87 | 81.05 |
| | FedMeta | 85.19 | 22.00 | 78.21 | 79.73 | 80.71 |
| | q-FFL | 85.26 | 16.89 | 78.50 | 79.94 | 81.00 |
| | CoPreFL ($\gamma = 0.75$) | **85.74** | **12.81** | **79.84** | **80.68** | **81.49** |
| Non-IID | FedAvg | 85.47 | 14.36 | 77.63 | 79.29 | 80.33 |
| | FedMeta | 85.74 | 17.64 | 77.92 | 79.80 | 81.10 |
| | q-FFL | 85.82 | 17.64 | 79.08 | 80.52 | 81.58 |
| | CoPreFL ($\gamma = 0.0$) | **86.25** | **12.96** | **79.87** | **80.99** | **81.68** |

Table 16: Average performance across 10 **IID** downstream FL tasks, initialized with various FL pre-trained methods using **25** out of 100 participants in **scenario I**, on the **Tiny-ImageNet** dataset.

| Pre-training (Scenario I, $|m| = 30$) | | Downstream: IID FL | | | | |
|---|---|---|---|---|---|---|
| Distribution | Method | Acc ↑ | Variance ↓ | Worst 10% ↑ | Worst 20% ↑ | Worst 30% ↑ |
| IID | FedAvg | 85.71 | **14.90** | 79.08 | 80.66 | 81.53 |
| | FedMeta | 85.48 | 15.92 | 79.51 | 80.16 | 81.19 |
| | q-FFL | 85.95 | 21.25 | 77.34 | 78.93 | 80.13 |
| | CoPreFL ($\gamma = 0.75$) | **86.05** | **14.90** | **80.31** | **81.49** | **82.27** |
| Non-IID | FedAvg | 85.64 | 21.07 | 75.90 | 77.99 | 79.32 |
| | FedMeta | 85.90 | 17.89 | 80.23 | 81.10 | 81.87 |
| | q-FFL | 86.49 | 14.75 | 78.79 | 80.16 | 81.24 |
| | CoPreFL ($\gamma = 0.0$) | **86.51** | **14.06** | **80.74** | **81.35** | **82.16** |

Table 17: Average performance across 10 **IID** downstream FL tasks, initialized with various FL pre-trained methods using **30** out of 100 participants in **scenario I**, on the **Tiny-ImageNet** dataset.

| Pre-training (Scenario I, $|m| = 15$) | | Downstream: Non-IID FL | | | | |
|---|---|---|---|---|---|---|
| Distribution | Method | Acc ↑ | Variance ↓ | Worst 10% ↑ | Worst 20% ↑ | Worst 30% ↑ |
| IID | FedAvg | 78.88 | 64.16 | 67.03 | 68.42 | 69.95 |
| | FedMeta | 82.62 | 43.16 | 70.76 | 73.48 | 74.48 |
| | q-FFL | 83.58 | 49.70 | 67.11 | 71.38 | 73.78 |
| | CoPreFL ($\gamma = 0.5$) | **83.83** | **41.22** | **73.28** | **74.39** | **75.50** |
| Non-IID | FedAvg | 82.19 | 38.32 | 72.64 | 73.90 | 75.38 |
| | FedMeta | 81.45 | 53.73 | 68.42 | 71.17 | 72.98 |
| | q-FFL | 82.85 | 32.26 | 73.89 | 76.14 | 77.28 |
| | CoPreFL ($\gamma = 0.25$) | **83.65** | **25.81** | **75.41** | **76.45** | **77.73** |

Table 18: Average performance across 10 **non-IID** downstream FL tasks, initialized with various FL pre-trained methods using **15** out of 100 participants in **scenario I**, on the **Tiny-ImageNet** dataset.

| Pre-training (Scenario I, $|m| = 20$) | | Downstream: Non-IID FL | | | | |
|---|---|---|---|---|---|---|
| Distribution | Method | Acc ↑ | Variance ↓ | Worst 10% ↑ | Worst 20% ↑ | Worst 30% ↑ |
| IID | FedAvg | 79.45 | 35.40 | 64.86 | 68.61 | 70.78 |
| | FedMeta | 81.68 | 65.61 | 65.96 | 69.17 | 71.59 |
| | q-FFL | 82.65 | 39.69 | 70.65 | 74.14 | 76.27 |
| | CoPreFL ($\gamma = 0.0$) | **83.79** | **34.93** | **72.59** | **75.05** | **76.76** |
| Non-IID | FedAvg | 82.94 | 37.21 | 68.99 | 72.29 | 74.40 |
| | FedMeta | 81.03 | 37.58 | 69.44 | 71.55 | 72.93 |
| | q-FFL | 84.11 | 43.96 | 73.87 | 76.05 | 77.37 |
| | CoPreFL ($\gamma = 0.5$) | **85.23** | **35.40** | **76.77** | **78.46** | **79.86** |

Table 19: Average performance across 10 **non-IID** downstream FL tasks, initialized with various FL pre-trained methods using **20** out of 100 participants in **scenario I**, on the **Tiny-ImageNet** dataset.

| Pre-training (Scenario I, $|m| = 25$) | | Downstream: Non-IID FL | | | | |
|---|---|---|---|---|---|---|
| Distribution | Method | Acc ↑ | Variance ↓ | Worst 10% ↑ | Worst 20% ↑ | Worst 30% ↑ |
| IID | FedAvg | 83.71 | 50.41 | 69.91 | 73.50 | 75.40 |
| | FedMeta | 84.19 | 42.90 | 73.77 | 76.22 | 77.77 |
| | q-FFL | 80.11 | 55.20 | 65.45 | 68.54 | 70.72 |
| | CoPreFL ($\gamma = 0.0$) | **84.29** | **36.60** | **76.02** | **77.56** | **78.95** |
| Non-IID | FedAvg | 79.08 | 55.80 | 66.80 | 69.06 | 71.38 |
| | FedMeta | 81.58 | 38.07 | 70.86 | 72.83 | 74.39 |
| | q-FFL | 83.16 | 45.56 | 72.39 | 75.29 | 77.09 |
| | CoPreFL ($\gamma = 0.25$) | **83.87** | **25.60** | **75.16** | **76.87** | **78.05** |

Table 20: Average performance across 10 **non-IID** downstream FL tasks, initialized with various FL pre-trained methods using **25** out of 100 participants in **scenario I**, on the **Tiny-ImageNet** dataset.

| Pre-training (Scenario I, $|m| = 30$) | | Downstream: Non-IID FL | | | | |
|---|---|---|---|---|---|---|
| Distribution | Method | Acc ↑ | Variance ↓ | Worst 10% ↑ | Worst 20% ↑ | Worst 30% ↑ |
| IID | FedAvg | 80.37 | 43.56 | 69.27 | 70.91 | 72.44 |
| | FedMeta | 80.51 | 44.09 | 68.05 | 70.74 | 72.18 |
| | q-FFL | 81.89 | 45.97 | 68.85 | 72.07 | 73.99 |
| | CoPreFL ($\gamma = 0.0$) | **83.17** | **31.81** | **71.16** | **73.64** | **75.49** |
| Non-IID | FedAvg | 82.73 | 42.51 | 72.90 | 74.84 | 76.50 |
| | FedMeta | 82.58 | 34.81 | 71.67 | 74.39 | 75.85 |
| | q-FFL | 83.39 | 38.07 | 72.60 | 74.97 | 76.57 |
| | CoPreFL ($\gamma = 0.75$) | **84.25** | **30.11** | **76.18** | **77.54** | **78.73** |

Table 21: Average performance across 10 **non-IID** downstream FL tasks, initialized with various FL pre-trained methods using **30** out of 100 participants in **scenario I**, on the **Tiny-ImageNet** dataset.

| Pre-training (Scenario II, $|m| = 15$) | | Downstream: IID FL | | | | |
|---|---|---|---|---|---|---|
| Distribution | Method | Acc ↑ | Variance ↓ | Worst 10% ↑ | Worst 20% ↑ | Worst 30% ↑ |
| IID | FedAvg | 87.02 | 12.82 | 82.42 | 83.33 | 84.04 |
| | FedMeta | 87.07 | 10.76 | 81.94 | 82.73 | 83.43 |
| | q-FFL | 87.27 | 13.69 | 81.21 | 82.30 | 82.95 |
| | CoPreFL-SGD ($\gamma = 0.25$) | 87.87 | 13.32 | 82.73 | 83.58 | 84.06 |
| | CoPreFL ($\gamma = 0.25$) | **88.58** | **8.70** | **83.39** | **83.88** | **84.69** |
| Non-IID | FedAvg | 86.22 | 15.44 | 79.15 | 80.36 | 81.54 |
| | FedMeta | 86.09 | 11.42 | 80.12 | 81.03 | 81.98 |
| | q-FFL | 86.56 | 15.29 | 78.42 | 80.00 | 81.37 |
| | CoPreFL-SGD ($\gamma = 0.75$) | 86.73 | 12.46 | 80.61 | 81.76 | 82.63 |
| | CoPreFL ($\gamma = 0.75$) | **87.42** | **9.06** | **81.21** | **82.30** | **82.95** |

Table 22: Average performance across 10 **IID** downstream FL tasks, initialized with various FL pre-trained methods using **15** out of 100 participants in **scenario II**, on the **CIFAR-100** dataset.

| Pre-training (Scenario II, $|m| = 20$) | | Downstream: IID FL | | | | |
|---|---|---|---|---|---|---|
| Distribution | Method | Acc ↑ | Variance ↓ | Worst 10% ↑ | Worst 20% ↑ | Worst 30% ↑ |
| IID | FedAvg | 87.28 | 15.21 | 80.00 | 81.21 | 82.10 |
| | FedMeta | 87.27 | 12.46 | 81.45 | 82.12 | 82.95 |
| | q-FFL | 86.84 | 12.74 | 80.73 | 82.12 | 82.87 |
| | CoPreFL-SGD ($\gamma = 0.5$) | 87.67 | 12.32 | 81.82 | 83.09 | 83.92 |
| | CoPreFL ($\gamma = 0.5$) | **88.10** | **9.30** | **83.52** | **84.30** | **85.05** |
| Non-IID | FedAvg | 86.39 | 17.31 | 79.64 | 80.79 | 81.78 |
| | FedMeta | 86.32 | 12.46 | 80.61 | 81.45 | 82.26 |
| | q-FFL | 86.17 | 16.24 | 79.27 | 81.09 | 82.02 |
| | CoPreFL-SGD ($\gamma = 0.25$) | 86.63 | 11.76 | 81.21 | 82.00 | 82.46 |
| | CoPreFL ($\gamma = 0.25$) | **87.02** | **10.50** | **81.70** | **82.42** | **83.23** |

Table 23: Average performance across 10 **IID** downstream FL tasks, initialized with various FL pre-trained methods using **20** out of 100 participants in **scenario II**, on the **CIFAR-100** dataset.

| Pre-training (Scenario II, $|m| = 25$) | | Downstream: IID FL | | | | |
|---|---|---|---|---|---|---|
| Distribution | Method | Acc ↑ | Variance ↓ | Worst 10% ↑ | Worst 20% ↑ | Worst 30% ↑ |
| IID | FedAvg | 87.31 | 14.90 | 79.39 | 81.64 | 82.75 |
| | FedMeta | 86.81 | 10.89 | 80.97 | 82.30 | 83.07 |
| | q-FFL | 87.36 | 17.47 | 80.73 | 81.76 | 82.87 |
| | CoPreFL-SGD ($\gamma = 0.5$) | 87.98 | 11.22 | 82.55 | 83.15 | 83.92 |
| | CoPreFL ($\gamma = 0.5$) | **88.67** | **9.98** | **83.88** | **84.55** | **85.29** |
| Non-IID | FedAvg | 86.37 | 15.44 | 78.79 | 80.12 | 81.25 |
| | FedMeta | 85.49 | 16.89 | 79.27 | 80.36 | 81.25 |
| | q-FFL | 85.67 | 17.06 | 80.61 | 81.45 | 81.98 |
| | CoPreFL-SGD ($\gamma = 0.75$) | 86.40 | 13.10 | 80.62 | 81.45 | 82.34 |
| | CoPreFL ($\gamma = 0.75$) | **87.32** | **11.22** | **82.42** | **83.27** | **83.84** |

Table 24: Average performance across 10 **IID** downstream FL tasks, initialized with various FL pre-trained methods using **25** out of 100 participants in **scenario II**, on the **CIFAR-100** dataset.

| Pre-training (Scenario II, $|m| = 30$) | | Downstream: IID FL | | | | |
|---|---|---|---|---|---|---|
| Distribution | Method | Acc ↑ | Variance ↓ | Worst 10% ↑ | Worst 20% ↑ | Worst 30% ↑ |
| IID | FedAvg | 87.51 | 13.76 | 82.55 | 83.58 | 84.20 |
| | FedMeta | 87.25 | 12.39 | 81.70 | 82.55 | 83.03 |
| | q-FFL | 86.78 | 13.76 | 81.21 | 82.06 | 82.79 |
| | CoPreFL-SGD ($\gamma = 0.75$) | 87.75 | 13.40 | 81.52 | 82.94 | 83.70 |
| | CoPreFL ($\gamma = 0.75$) | **88.27** | **9.06** | **84.06** | **84.55** | **85.05** |
| Non-IID | FedAvg | 86.07 | 11.09 | 80.61 | 81.64 | 82.38 |
| | FedMeta | 86.25 | 12.96 | 79.03 | 80.36 | 81.66 |
| | q-FFL | 85.50 | 15.29 | 77.58 | 79.39 | 80.57 |
| | CoPreFL-SGD ($\gamma = 0.5$) | 86.47 | **10.96** | 80.36 | 81.36 | 82.00 |
| | CoPreFL ($\gamma = 0.5$) | **87.54** | **10.96** | **81.09** | **81.67** | **82.40** |

Table 25: Average performance across 10 **IID** downstream FL tasks, initialized with various FL pre-trained methods using **30** out of 100 participants in **scenario II**, on the **CIFAR-100** dataset.

| Pre-training (Scenario II, $|m| = 15$) | | Downstream: Non-IID FL | | | | |
|---|---|---|---|---|---|---|
| Distribution | Method | Acc ↑ | Variance ↓ | Worst 10% ↑ | Worst 20% ↑ | Worst 30% ↑ |
| IID | FedAvg | 84.02 | 46.79 | 71.01 | 74.53 | 76.81 |
| | FedMeta | 83.47 | 34.11 | 73.68 | 75.20 | 76.35 |
| | q-FFL | 85.03 | 35.64 | 74.04 | 76.39 | 78.12 |
| | CoPreFL-SGD ($\gamma = 0.75$) | 85.04 | 35.64 | **74.61** | 76.40 | 78.34 |
| | CoPreFL ($\gamma = 0.75$) | **85.08** | **31.70** | **74.61** | **76.87** | **78.63** |
| Non-IID | FedAvg | 82.91 | 41.99 | 71.90 | 75.23 | 76.81 |
| | FedMeta | 78.77 | 70.39 | 65.13 | 67.47 | 69.28 |
| | q-FFL | 80.94 | 49.42 | 69.57 | 71.46 | 72.86 |
| | CoPreFL-SGD ($\gamma = 0.25$) | 83.42 | 40.20 | 73.09 | 74.54 | 76.29 |
| | CoPreFL ($\gamma = 0.25$) | **83.83** | **39.31** | **74.26** | **76.42** | **78.10** |

Table 26: Average performance across 10 **non-IID** downstream FL tasks, initialized with various FL pre-trained methods using **15** out of 100 participants in **scenario II**, on the **CIFAR-100** dataset.

| Pre-training (Scenario II, $|m| = 20$) | | Downstream: Non-IID FL | | | | |
|---|---|---|---|---|---|---|
| Distribution | Method | Acc ↑ | Variance ↓ | Worst 10% ↑ | Worst 20% ↑ | Worst 30% ↑ |
| IID | FedAvg | 81.79 | 41.73 | 69.84 | 73.47 | 75.11 |
| | FedMeta | 82.29 | 47.75 | 71.69 | 74.17 | 75.71 |
| | q-FFL | 82.40 | 40.32 | 73.96 | 75.30 | 76.59 |
| | CoPreFL-SGD ($\gamma = 0.75$) | 82.90 | 38.94 | 73.02 | 75.60 | 77.18 |
| | CoPreFL ($\gamma = 0.75$) | **85.68** | **27.14** | **75.36** | **77.25** | **78.49** |
| Non-IID | FedAvg | 82.82 | 49.00 | 69.71 | 72.54 | 74.58 |
| | FedMeta | 82.69 | 48.44 | 68.84 | 71.82 | 74.14 |
| | q-FFL | 82.14 | 73.10 | 68.22 | 70.64 | 73.77 |
| | CoPreFL-SGD ($\gamma = 0.25$) | 83.63 | 41.73 | 69.76 | 73.46 | 75.64 |
| | CoPreFL ($\gamma = 0.25$) | **86.63** | **31.58** | **73.05** | **75.82** | **77.58** |

Table 27: Average performance across 10 **non-IID** downstream FL tasks, initialized with various FL pre-trained methods using **20** out of 100 participants in **scenario II**, on the **CIFAR-100** dataset.

| Pre-training (Scenario II, $|m| = 25$) | | Downstream: Non-IID FL | | | | |
|---|---|---|---|---|---|---|
| Distribution | Method | Acc ↑ | Variance ↓ | Worst 10% ↑ | Worst 20% ↑ | Worst 30% ↑ |
| IID | FedAvg | 80.53 | 62.57 | 66.51 | 68.54 | 70.78 |
| | FedMeta | 82.37 | 45.97 | 70.68 | 73.40 | 75.21 |
| | q-FFL | 82.06 | 48.44 | 71.08 | 73.03 | 74.71 |
| | CoPreFL-SGD ($\gamma = 0.25$) | 82.62 | 75.86 | 68.12 | 70.73 | 72.51 |
| | CoPreFL ($\gamma = 0.25$) | **85.05** | **33.99** | **75.12** | **76.74** | **77.79** |
| Non-IID | FedAvg | 84.06 | 40.07 | 71.11 | 73.36 | 75.67 |
| | FedMeta | 81.40 | 47.33 | 67.41 | 70.87 | 72.49 |
| | q-FFL | 82.30 | 55.06 | 67.82 | 71.53 | 73.70 |
| | CoPreFL-SGD ($\gamma = 0.5$) | 84.25 | 53.88 | 71.62 | 73.48 | 75.92 |
| | CoPreFL ($\gamma = 0.5$) | **84.92** | **39.82** | **75.04** | **77.45** | **78.93** |

Table 28: Average performance across 10 **non-IID** downstream FL tasks, initialized with various FL pre-trained methods using **25** out of 100 participants in **scenario II**, on the **CIFAR-100** dataset.

| Pre-training (Scenario II, $|m| = 30$) | | Downstream: Non-IID FL | | | | |
|---|---|---|---|---|---|---|
| Distribution | Method | Acc ↑ | Variance ↓ | Worst 10% ↑ | Worst 20% ↑ | Worst 30% ↑ |
| IID | FedAvg | 82.70 | 62.09 | 66.99 | 71.18 | 73.18 |
| | FedMeta | 83.00 | 39.94 | 71.16 | 73.43 | 75.52 |
| | q-FFL | 82.81 | 44.09 | 71.82 | 73.68 | 75.31 |
| | CoPreFL-SGD ($\gamma = 0.25$) | 85.05 | 37.33 | 75.16 | 76.79 | 78.21 |
| | CoPreFL ($\gamma = 0.25$) | **85.78** | **35.88** | **75.26** | **78.60** | **80.55** |
| Non-IID | FedAvg | 81.14 | 71.23 | 65.42 | 69.17 | 70.99 |
| | FedMeta | 78.98 | 64.48 | 63.97 | 66.89 | 69.06 |
| | q-FFL | 79.87 | 70.06 | 63.96 | 67.47 | 70.16 |
| | CoPreFL-SGD ($\gamma = 0.75$) | 83.21 | 37.94 | 72.75 | 74.53 | 76.01 |
| | CoPreFL ($\gamma = 0.75$) | **85.11** | **36.84** | **72.66** | **75.63** | **77.47** |

Table 29: Average performance across 10 **non-IID** downstream FL tasks, initialized with various FL pre-trained methods using **30** out of 100 participants in **scenario II**, on the **CIFAR-100** dataset.

| Pre-training (Scenario II, $|m| = 15$) | | Downstream: IID FL | | | | |
|---|---|---|---|---|---|---|
| Distribution | Method | Acc ↑ | Variance ↓ | Worst 10% ↑ | Worst 20% ↑ | Worst 30% ↑ |
| IID | FedAvg | 85.79 | 16.16 | 77.34 | 78.37 | 80.17 |
| | FedMeta | 85.88 | 17.47 | 77.49 | 78.93 | 80.33 |
| | q-FFL | 85.24 | 15.60 | 77.38 | 78.37 | 80.23 |
| | CoPreFL-SGD ($\gamma = 0.75$) | 85.37 | 14.82 | 77.49 | 79.00 | 80.33 |
| | CoPreFL ($\gamma = 0.75$) | **86.64** | **14.59** | **80.23** | **81.17** | **82.06** |
| Non-IID | FedAvg | 85.17 | 16.56 | 78.21 | 79.73 | 80.52 |
| | FedMeta | 85.76 | 18.40 | 78.93 | 80.52 | 81.39 |
| | q-FFL | 86.29 | 18.06 | 78.79 | 80.38 | 81.58 |
| | CoPreFL-SGD ($\gamma = 0.25$) | 85.49 | 13.84 | 79.65 | 80.66 | 82.07 |
| | CoPreFL ($\gamma = 0.25$) | **86.68** | **12.67** | **80.09** | **81.02** | **82.36** |

Table 30: Average performance across 10 **IID** downstream FL tasks, initialized with various FL pre-trained methods using **15** out of 100 participants in **scenario II**, on the **Tiny-ImageNet** dataset.

| Pre-training (Scenario II, $|m| = 20$) | | Downstream: IID FL | | | | |
|---|---|---|---|---|---|---|
| Distribution | Method | Acc ↑ | Variance ↓ | Worst 10% ↑ | Worst 20% ↑ | Worst 30% ↑ |
| IID | FedAvg | 85.08 | 14.29 | 78.21 | 79.44 | 80.52 |
| | FedMeta | 85.39 | 19.89 | 78.33 | 79.30 | 80.48 |
| | q-FFL | 85.41 | 20.70 | 77.63 | 79.22 | 80.28 |
| | CoPreFL-SGD ($\gamma = 0.75$) | 85.57 | 17.89 | 78.64 | 80.01 | 80.95 |
| | CoPreFL ($\gamma = 0.75$) | **86.77** | **12.25** | **80.52** | **81.17** | **81.96** |
| Non-IID | FedAvg | 85.15 | 20.98 | 79.04 | 80.45 | 81.34 |
| | FedMeta | 85.38 | 14.82 | 78.79 | 80.59 | 81.58 |
| | q-FFL | 85.46 | 19.71 | 78.81 | 80.11 | 81.97 |
| | CoPreFL-SGD ($\gamma = 0.75$) | 85.57 | 18.75 | 79.65 | 81.10 | 82.06 |
| | CoPreFL ($\gamma = 0.75$) | **86.74** | **12.82** | **80.66** | **81.60** | **82.49** |

Table 31: Average performance across 10 **IID** downstream FL tasks, initialized with various FL pre-trained methods using **20** out of 100 participants in **scenario II**, on the **Tiny-ImageNet** dataset.

| Pre-training (Scenario II, $|m| = 25$) | | Downstream: IID FL | | | | |
|---|---|---|---|---|---|---|
| Distribution | Method | Acc ↑ | Variance ↓ | Worst 10% ↑ | Worst 20% ↑ | Worst 30% ↑ |
| IID | FedAvg | 85.99 | 16.65 | 78.50 | 79.73 | 80.66 |
| | FedMeta | 85.58 | 19.89 | 78.21 | 79.80 | 80.81 |
| | q-FFL | 85.66 | 17.22 | 78.07 | 79.73 | 80.71 |
| | CoPreFL-SGD ($\gamma = 0.5$) | 86.33 | 15.44 | 80.63 | 81.35 | **82.28** |
| | CoPreFL ($\gamma = 0.5$) | **86.72** | **15.29** | **80.94** | **81.59** | 82.20 |
| Non-IID | FedAvg | 85.50 | 16.48 | 78.35 | 79.80 | 80.86 |
| | FedMeta | 86.57 | 17.81 | 78.93 | 79.80 | 80.86 |
| | q-FFL | 86.45 | 14.82 | 79.08 | 80.74 | 81.96 |
| | CoPreFL-SGD ($\gamma = 0.5$) | 86.61 | 13.62 | 79.37 | 80.45 | 81.19 |
| | CoPreFL ($\gamma = 0.5$) | **87.16** | **10.43** | **80.38** | **81.39** | **82.20** |

Table 32: Average performance across 10 **IID** downstream FL tasks, initialized with various FL pre-trained methods using **25** out of 100 participants in **scenario II**, on the **Tiny-ImageNet** dataset.

| Pre-training (Scenario II, $|m| = 30$) | | Downstream: IID FL | | | | |
|---|---|---|---|---|---|---|
| Distribution | Method | Acc ↑ | Variance ↓ | Worst 10% ↑ | Worst 20% ↑ | Worst 30% ↑ |
| IID | FedAvg | 85.27 | 14.90 | 77.92 | 79.15 | 80.04 |
| | FedMeta | 85.61 | 13.54 | 78.07 | 79.87 | 81.14 |
| | q-FFL | 85.34 | 17.22 | 80.37 | 81.39 | 82.15 |
| | CoPreFL-SGD ($\gamma = 1.0$) | 85.61 | 12.25 | 79.94 | 80.74 | 81.29 |
| | CoPreFL ($\gamma = 1.0$) | **86.62** | **11.69** | **81.24** | **81.89** | **82.64** |
| Non-IID | FedAvg | 85.70 | 20.61 | 78.64 | 80.09 | 81.19 |
| | FedMeta | 85.44 | 16.56 | 79.08 | 80.38 | 81.34 |
| | q-FFL | 85.54 | 17.56 | 79.22 | 80.66 | 81.67 |
| | CoPreFL-SGD ($\gamma = 0.5$) | 85.79 | 14.21 | 79.65 | 80.59 | 81.43 |
| | CoPreFL ($\gamma = 0.5$) | **86.15** | **13.25** | **80.66** | **81.89** | **82.68** |

Table 33: Average performance across 10 **IID** downstream FL tasks, initialized with various FL pre-trained methods using **30** out of 100 participants in **scenario II**, on the **Tiny-ImageNet** dataset.

| Pre-training (Scenario II, $|m| = 15$) | | Downstream: Non-IID FL | | | | |
|---|---|---|---|---|---|---|
| **Distribution** | **Method** | **Acc ↑** | **Variance ↓** | **Worst 10% ↑** | **Worst 20% ↑** | **Worst 30% ↑** |
| | FedAvg | 82.50 | 39.06 | 70.33 | 73.19 | 75.15 |
| | FedMeta | 81.57 | 62.57 | 67.65 | 70.95 | 72.83 |
| IID | q-FFL | 82.31 | 48.16 | 70.00 | 72.07 | 73.99 |
| | CoPreFL-SGD ($\gamma = 0.75$) | 83.11 | 38.94 | 72.06 | 73.63 | 75.29 |
| | CoPreFL ($\gamma = 0.75$) | **84.68** | **33.76** | **73.84** | **75.50** | **77.35** |
| | FedAvg | 80.19 | 53.00 | 67.13 | 69.67 | 71.36 |
| | FedMeta | 81.94 | 56.40 | 67.15 | 71.39 | 73.39 |
| Non-IID | q-FFL | 81.64 | 54.46 | 69.58 | 71.65 | 73.02 |
| | CoPreFL-SGD ($\gamma = 0.25$) | 83.57 | 41.22 | 71.46 | 73.45 | 75.41 |
| | CoPreFL ($\gamma = 0.25$) | **84.26** | **28.52** | **73.61** | **75.55** | **76.79** |

Table 34: Average performance across 10 **non-IID** downstream FL tasks, initialized with various FL pre-trained methods using **15** out of 100 participants in **scenario II**, on the **Tiny-ImageNet** dataset.

| Pre-training (Scenario II, $|m| = 20$) | | Downstream: Non-IID FL | | | | |
|---|---|---|---|---|---|---|
| **Distribution** | **Method** | **Acc ↑** | **Variance ↓** | **Worst 10% ↑** | **Worst 20% ↑** | **Worst 30% ↑** |
| | FedAvg | 81.91 | 77.97 | 64.37 | 71.67 | 74.51 |
| | FedMeta | 81.58 | 38.56 | 70.94 | 71.82 | 73.09 |
| IID | q-FFL | 82.17 | 48.58 | 70.22 | 72.66 | 74.24 |
| | CoPreFL-SGD ($\gamma = 0.5$) | 82.32 | 42.25 | 71.61 | 73.31 | 74.62 |
| | CoPreFL ($\gamma = 0.5$) | **84.48** | **35.64** | **73.66** | **74.75** | **76.21** |
| | FedAvg | 82.87 | 48.16 | 68.94 | 72.91 | 75.28 |
| | FedMeta | 84.19 | 49.70 | 70.41 | 72.63 | 74.74 |
| Non-IID | q-FFL | 83.51 | 44.22 | 69.91 | 73.71 | 76.01 |
| | CoPreFL-SGD ($\gamma = 0.5$) | 84.30 | 36.24 | 72.83 | 75.64 | 77.37 |
| | CoPreFL ($\gamma = 0.5$) | **84.72** | **24.80** | **75.84** | **77.31** | **78.50** |

Table 35: Average performance across 10 **non-IID** downstream FL tasks, initialized with various FL pre-trained methods using **20** out of 100 participants in **scenario II**, on the **Tiny-ImageNet** dataset.

| Pre-training (Scenario II, $|m| = 25$) | | Downstream: Non-IID FL | | | | |
|---|---|---|---|---|---|---|
| **Distribution** | **Method** | **Acc ↑** | **Variance ↓** | **Worst 10% ↑** | **Worst 20% ↑** | **Worst 30% ↑** |
| | FedAvg | 83.42 | 44.36 | 70.04 | 72.88 | 75.10 |
| | FedMeta | 80.66 | 45.29 | 68.38 | 70.89 | 73.05 |
| IID | q-FFL | 83.60 | 34.93 | 74.41 | 75.86 | 77.29 |
| | CoPreFL-SGD ($\gamma = 0.75$) | 84.49 | 46.51 | 72.66 | 74.48 | 75.97 |
| | CoPreFL ($\gamma = 0.75$) | **84.81** | **32.15** | **76.65** | **77.97** | **79.12** |
| | FedAvg | 79.44 | 64.80 | 65.35 | 68.25 | 70.19 |
| | FedMeta | 81.22 | 48.58 | 69.53 | 71.87 | 73.52 |
| Non-IID | q-FFL | 82.14 | 41.60 | 72.32 | 74.56 | 76.31 |
| | CoPreFL-SGD ($\gamma = 0.5$) | 82.88 | 35.64 | 71.25 | 73.00 | 74.32 |
| | CoPreFL ($\gamma = 0.5$) | **84.02** | **24.01** | **75.49** | **77.02** | **78.17** |

Table 36: Average performance across 10 **non-IID** downstream FL tasks, initialized with various FL pre-trained methods using **25** out of 100 participants in **scenario II**, on the **Tiny-ImageNet** dataset.

| Pre-training (Scenario II, $|m| = 30$) | | Downstream: Non-IID FL | | | | |
|---|---|---|---|---|---|---|
| **Distribution** | **Method** | **Acc ↑** | **Variance ↓** | **Worst 10% ↑** | **Worst 20% ↑** | **Worst 30% ↑** |
| | FedAvg | 83.63 | 42.25 | 71.27 | 73.56 | 75.00 |
| | FedMeta | 83.41 | 39.82 | 72.34 | 72.81 | 73.15 |
| IID | q-FFL | 82.98 | 63.68 | 62.50 | 65.62 | 68.31 |
| | CoPreFL-SGD ($\gamma = 0.25$) | 83.66 | 42.90 | 71.84 | 74.30 | 75.70 |
| | CoPreFL ($\gamma = 0.25$) | **84.26** | **39.31** | **73.77** | **75.91** | **77.58** |
| | FedAvg | 83.24 | 42.25 | 69.06 | 73.39 | 75.31 |
| | FedMeta | 81.61 | 47.89 | 69.38 | 72.47 | 74.39 |
| Non-IID | q-FFL | 81.92 | 51.55 | 69.73 | 72.17 | 74.38 |
| | CoPreFL-SGD ($\gamma = 0.0$) | 83.37 | 46.38 | 72.17 | 74.15 | 75.53 |
| | CoPreFL ($\gamma = 0.0$) | **85.45** | **38.32** | **74.43** | **75.90** | **77.45** |

Table 37: Average performance across 10 **non-IID** downstream FL tasks, initialized with various FL pre-trained methods using **30** out of 100 participants in **scenario II**, on the **Tiny-ImageNet** dataset.

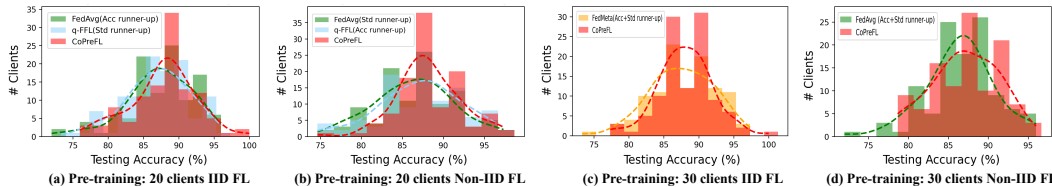

Figure 3: The distributions of testing accuracy in **IID FL** downstream tasks under various pre-training setups in **scenario I** on the **CIFAR-100** dataset.

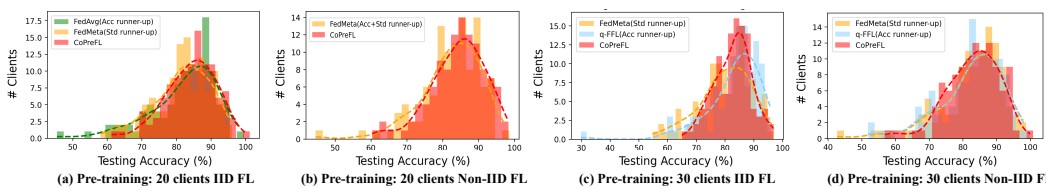

Figure 4: The distributions of testing accuracy in **non-IID FL** downstream tasks under various pre-training setups in **scenario I** on the **CIFAR-100** dataset.

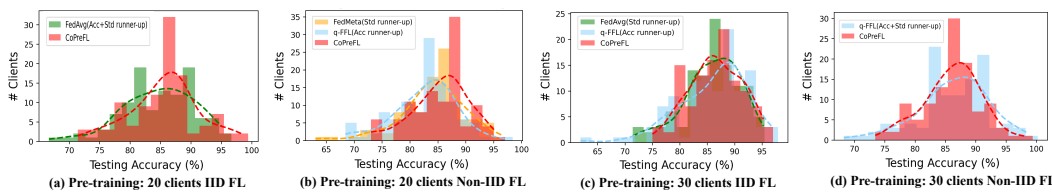

Figure 5: The distributions of testing accuracy in **IID FL** downstream tasks under various pre-training setups in **scenario I** on the **Tiny-ImageNet** dataset.

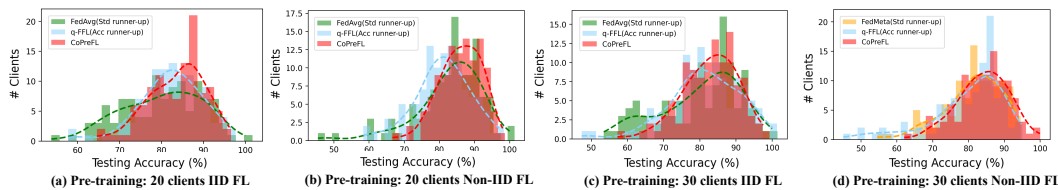

Figure 6: The distributions of testing accuracy in **non-IID FL** downstream tasks under various pre-training setups in **scenario I** on the **Tiny-ImageNet** dataset

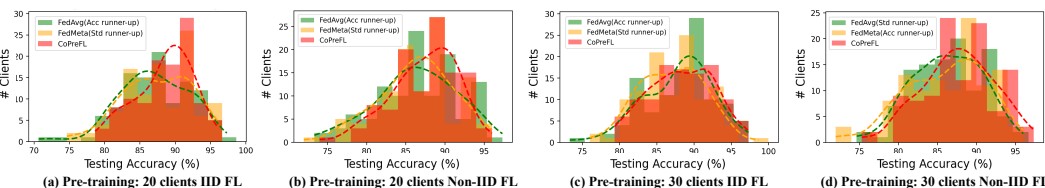

Figure 7: The distributions of testing accuracy in **IID FL** downstream tasks under various pre-training setups in **scenario II** on the **CIFAR-100** dataset.

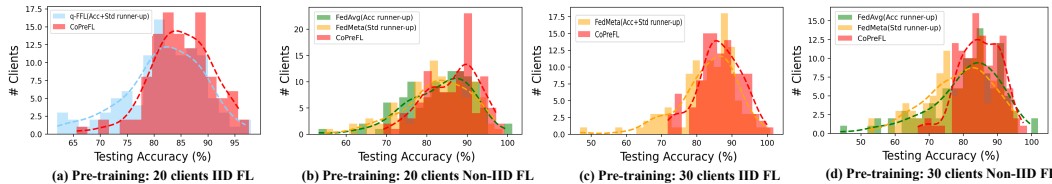

Figure 8: The distributions of testing accuracy in **non-IID FL** downstream tasks under various pre-training setups in **scenario II** on the **CIFAR-100** dataset

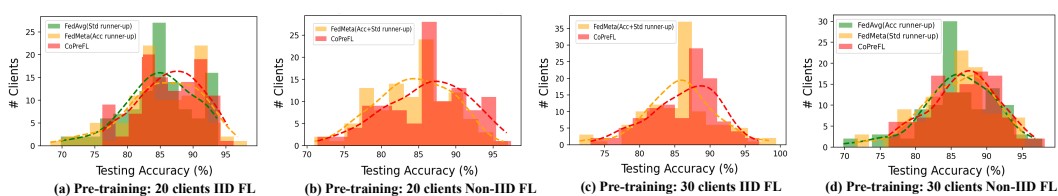

Figure 9: The distributions of testing accuracy in **IID FL** downstream tasks under various pre-training setups in **scenario II** on the **Tiny-ImageNet** dataset.

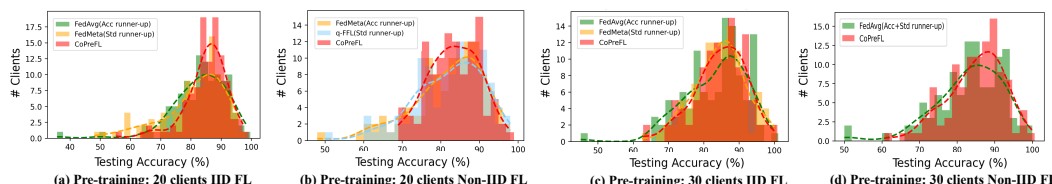

Figure 10: The distributions of testing accuracy in **non-IID FL** downstream tasks under various pre-training setups in **scenario II** on the **Tiny-ImageNet** dataset.

| Pre-training | | Downstream: IID FL | |
|---|---|---|---|
| Scenario | Method | Acc ↑ | Variance ↓ |
| I | Random | 78.03 | 16.17 |
| | Centralized | 83.17 | 17.93 |
| | CoPreFL | **86.32** | **14.14** |
| II | Random | 78.21 | 16.44 |
| | Centralized | 84.39 | 15.92 |
| | CoPreFL | **87.02** | **10.50** |

Table 38: Average performance of IID FL tasks, initialized by different methods pre-trained in two scenarios, on CIFAR-100.

| Pre-training | | Downstream: IID FL | |
|---|---|---|---|
| Scenario | Method | Acc ↑ | Variance ↓ |
| I | Random | 81.29 | 18.33 |
| | Centralized | 83.81 | 19.03 |
| | CoPreFL | **86.00** | **16.16** |
| II | Random | 83.16 | 16.08 |
| | Centralized | 84.36 | 17.89 |
| | CoPreFL | **86.74** | **12.82** |

Table 39: Average performance of IID FL tasks, initialized by different methods pre-trained in two scenarios, on Tiny-ImageNet.

| Pre-training | | Downstream: non-IID FL | |
|---|---|---|---|
| Scenario | Method | Acc ↑ | Variance ↓ |
| I | Random | 75.50 | 54.88 |
| | Centralized | 83.19 | 42.69 |
| | CoPreFL | **85.23** | **35.40** |
| II | Random | 76.23 | 61.62 |
| | Centralized | 82.39 | 39.31 |
| | CoPreFL | **84.72** | **24.80** |

Table 40: Average performance of non-IID FL tasks, initialized by different methods pre-trained in two scenarios, on Tiny-ImageNet.

| Pre-training | Downstream: non-IID FL | |
|---|---|---|
| Method | Acc ↑ | Variance ↓ |
| Centralized | 86.75 | 67.34 |
| CoPreFL | 87.96 | 30.79 |

Table 41: Average performance of non-IID FL tasks, initialized by different pre-trained models. Note that ImageNet is used for pre-training, while CIFAR-100 is used for downstream FL.

| **Pre-training** (Scenario I, $|m| = 20$) | | **Downstream:** Non-IID FL | | | | |
|---|---|---|---|---|---|---|
| **Distribution** | **Method** | **Acc** ↑ | **Variance** ↓ | **Worst 10%** ↑ | **Worst 20%** ↑ | **Worst 30%** ↑ |
| IID | FedDyn | 83.15 | 40.13 | 69.55 | 71.32 | 73.95 |
| | CoPreFL ($\gamma = 0.25$) | **84.36** | **38.56** | **73.66** | **75.63** | **77.40** |
| Non-IID | FedDyn | 81.23 | 53.17 | 68.39 | 70.23 | 72.46 |
| | CoPreFL ($\gamma = 0.75$) | **83.29** | **34.69** | **71.58** | **73.20** | **74.59** |

Table 42: Comparison with Non-IID related method.

| **Pre-training** (Scenario I, $|m| = 20$) | | **Downstream:** IID FL | | | | |
|---|---|---|---|---|---|---|
| **Distribution** | **Method** | **Acc** ↑ | **Variance** ↓ | **Worst 10%** ↑ | **Worst 20%** ↑ | **Worst 30%** ↑ |
| IID | FedAvg | 86.84 | 2.43 | 76.88 | 79.87 | 81.47 |
| | FedMeta | 82.16 | 2.31 | 75.32 | 75.97 | 76.62 |
| | q-FFL | 79.91 | 3.09 | 76.62 | 77.40 | 77.83 |
| | CoPreFL ($\gamma = 0.75$) | **91.59** | **1.61** | **86.75** | **87.40** | **88.31** |
| Non-IID | FedAvg | 83.84 | 2.13 | 75.97 | 77.88 | 80.79 |
| | FedMeta | 86.71 | 1.61 | 74.69 | 77.82 | 78.21 |
| | q-FFL | 79.85 | 2.53 | 69.22 | 71.95 | 74.94 |
| | CoPreFL ($\gamma = 0.5$) | **89.01** | **1.46** | **81.29** | **83.55** | **84.24** |

Table 43: Average performance across 10 **IID** downstream FL tasks, initialized with various FL pre-trained methods using **20** out of 100 participants in **scenario I**, on the **FEMNIST** dataset.

| **Pre-training** (Scenario I, $|m| = 20$) | | **Downstream:** Non-IID FL | | | | |
|---|---|---|---|---|---|---|
| **Distribution** | **Method** | **Acc** ↑ | **Variance** ↓ | **Worst 10%** ↑ | **Worst 20%** ↑ | **Worst 30%** ↑ |
| IID | FedAvg | 75.04 | 16.16 | 70.75 | 71.65 | 72.19 |
| | FedMeta | 69.61 | 6.81 | 59.19 | 62.35 | 64.54 |
| | q-FFL | 70.47 | 19.30 | 58.31 | 61.66 | 63.67 |
| | CoPreFL ($\gamma = 0.5$) | **78.38** | **6.70** | **72.77** | **74.54** | **75.35** |
| Non-IID | FedAvg | 70.74 | 28.58 | 65.06 | 66.20 | 66.99 |
| | FedMeta | 64.02 | 33.29 | 60.91 | 61.57 | 62.07 |
| | q-FFL | 68.04 | 31.55 | 58.09 | 60.57 | 61.72 |
| | CoPreFL ($\gamma = 0.5$) | **72.65** | **24.89** | **67.49** | **68.40** | **69.32** |

Table 44: Average performance across 10 **Non-IID** downstream FL tasks, initialized with various FL pre-trained methods using **20** out of 100 participants in **scenario I**, on the **FEMNIST** dataset.

| **Pre-training** (Scenario II, $|m| = 20$) | | **Downstream:** IID FL | | | | |
|---|---|---|---|---|---|---|
| **Distribution** | **Method** | **Acc** ↑ | **Variance** ↓ | **Worst 10%** ↑ | **Worst 20%** ↑ | **Worst 30%** ↑ |
| IID | FedAvg | 86.70 | 6.86 | 75.24 | 77.94 | 80.58 |
| | FedMeta | 81.29 | 3.42 | 70.82 | 72.10 | 74.87 |
| | q-FFL | 87.07 | 2.85 | 79.16 | 83.52 | 83.72 |
| | CoPreFL-SGD ($\gamma = 0.5$) | 86.31 | 5.33 | 77.16 | 79.72 | 81.23 |
| | CoPreFL ($\gamma = 0.5$) | **90.33** | **2.22** | **82.03** | **83.54** | **84.09** |
| Non-IID | FedAvg | 85.24 | 7.78 | 77.01 | 79.25 | 80.52 |
| | FedMeta | 83.52 | 5.76 | 71.44 | 75.37 | 77.76 |
| | q-FFL | 87.11 | 10.24 | 74.83 | 75.52 | 76.44 |
| | CoPreFL-SGD ($\gamma = 0.25$) | 87.05 | 11.22 | 73.30 | 76.42 | 79.76 |
| | CoPreFL ($\gamma = 0.25$) | **89.01** | **5.47** | **79.63** | **81.22** | **82.71** |

Table 45: Average performance across 10 **IID** downstream FL tasks, initialized with various FL pre-trained methods using **20** out of 100 participants in **scenario II**, on the **FEMNIST** dataset.

| Pre-training (Scenario I, $|m| = 20$) | | Downstream: Non-IID FL | | | | |
|---|---|---|---|---|---|---|
| Distribution | Method | Acc ↑ | Variance ↓ | Worst 10% ↑ | Worst 20% ↑ | Worst 30% ↑ |
| IID | FedAvg | 71.25 | 15.28 | 54.99 | 58.71 | 60.31 |
| | FedMeta | 73.29 | 16.89 | 61.39 | 65.22 | 66.17 |
| | q-FFL | 77.93 | 8.88 | 65.91 | 66.74 | 68.03 |
| | CoPreFL-SGD ($\gamma = 0.0$) | 76.19 | 9.30 | 66.07 | 66.62 | 67.80 |
| | CoPreFL ($\gamma = 0.0$) | **82.33** | **7.95** | **68.31** | **70.37** | **72.19** |
| Non-IID | FedAvg | 66.31 | 21.06 | 44.79 | 50.93 | 53.29 |
| | FedMeta | 71.49 | 13.10 | 58.31 | 59.27 | 61.33 |
| | q-FFL | 74.99 | 29.26 | 61.20 | 63.98 | 65.01 |
| | CoPreFL-SGD ($\gamma = 0.75$) | 72.66 | 29.05 | 58.71 | 61.29 | 63.32 |
| | CoPreFL ($\gamma = 0.75$) | **79.31** | **9.55** | **63.29** | **65.33** | **66.92** |

Table 46: Average performance across 10 **Non-IID** downstream FL tasks, initialized with various FL pre-trained methods using **20** out of 100 participants in **scenario II**, on the **FEMNIST** dataset.

| Pre-training (Scenario I) | Downstream: Non-IID FL | | | | |
|---|---|---|---|---|---|
| Method | Acc ↑ | Variance ↓ | Worst 10% ↑ | Worst 20% ↑ | Worst 30% ↑ |
| Centralized | 82.63 | 63.57 | 67.35 | 69.22 | 70.38 |
| FedAvg | 80.19 | 51.35 | 68.72 | 70.15 | 72.33 |
| FedMeta | 83.14 | 39.85 | 67.29 | 71.35 | 73.81 |
| q-FFL | 81.34 | 47.98 | 69.22 | 70.35 | 72.37 |
| CoPreFL ($\gamma = 0.5$) | **84.79** | **30.51** | **70.83** | **72.66** | **74.10** |

Table 47: Average performance across 10 non-IID downstream FL tasks, initialized with centralized model and various non-IID FL pre-trained models, encompassing both seen and unseen classes.

| Pre-training (Scenario I) | Downstream: Non-IID FedProx | | | | |
|---|---|---|---|---|---|
| Method | Acc ↑ | Variance ↓ | Worst 10% ↑ | Worst 20% ↑ | Worst 30% ↑ |
| Centralized | 82.39 | 51.46 | 70.33 | 71.28 | 73.52 |
| FedAvg | 79.53 | 46.15 | 63.17 | 69.74 | 71.59 |
| FedMeta | 81.77 | 63.12 | 63.58 | 68.19 | 70.28 |
| q-FFL | 83.19 | 52.12 | 67.41 | 70.59 | 72.33 |
| CoPreFL ($\gamma = 0.25$) | **84.31** | **30.55** | 70.19 | **73.88** | **75.13** |

Table 48: Average performance across 10 non-IID downstream FedProx tasks, initialized with centralized model and various non-IID FL pre-trained models.

| Pre-training (Scenario I, $|m| = 20$) | | Downstream: Non-IID FL | | | | |
|---|---|---|---|---|---|---|
| Distribution | Method | Acc ↑ | Variance ↓ | Worst 10% ↑ | Worst 20% ↑ | Worst 30% ↑ |
| Non-IID | Per-FedAvg | 81.58 | 49.73 | 67.21 | 71.05 | 72.39 |
| | CoPreFL ($\gamma = 0.75$) | **83.29** | **34.69** | **71.58** | **73.20** | **74.59** |

Table 49: Comparison with personalized FL method.

