# OpenReview forum: "Rethinking the Starting Point: Enhancing Performance and Fairness of Federated Learning via Collaborative Pre-Training"
_ICLR.cc/2024/Conference — Submitted to ICLR 2024_

### Official Review · Reviewer_4hah · 2023-10-26

**Soundness:** 2 fair
**Presentation:** 3 good
**Contribution:** 3 good
**Rating:** 3
**Confidence:** 3

**Summary:**

This paper studies the topic of federated pretraining, and the authors propose a distributed pretraining method, which utilizes meta-learning to simulate the distributed scenario and also balances performance and fairness. The proposed method is evaluated on three public datasets and outperforms compared methods.

**Strengths:**

-	This paper studies an important topic. Exploring distributed pretraining strategies is helpful for downstream tasks.
-	The motivation for performing pretraining in a distributed way is clear.
-	The paper's organization is easy to follow.
-	The proposed method outperforms the compared methods on three datasets.

**Weaknesses:**

-	The observation of centralized pretraining causes side effects like performance biases are not well analyzed/explained.
-	The idea of using meta-learning to simulate distributed scenarios may not make sense for non-iid settings. If the clients' data are non-iid distributed, using local data for pretraining may not help. The local simulation cannot represent the global distributions.
-	The meta-learning method part is not clear. The data split needs to be clarified.
-	The technical contribution of applying meta-learning for a new distributed pretraining setting might be marginal.
-	Compared with using public datasets for local pretraining, this method suffers further communication costs.
-	The experimental settings of splitting classes into pretraining and unseen downstream tasks are not practical.
-	Some experiment details lack of explanation.

**Questions:**

-	Why does centralized pretraining cause side effects like performance biases? It is better to elaborate more on this.
-	The meta-learning part is not clear. How to split the unseen task data and the client data for pretraining? The assumption that the downstream tasks are unseen may not be proper enough. For example, if a client joins FL, it usually has specific goals, and the client data will be task-specific. The clients have no motivation to hold unrelated data for pretraining. And it is also weird to keep the task-specific (downstream) data unseen.
-	Following the previous point, the experimental settings are also not practical to me. First, clients in FL usually may not hold 'new classes' for pretraining tasks; the client data are strongly related to the task in general. Second, for real-world FL, client local data are valuable and cost a lot to annotate. It does not make sense to split out part of classes for pre-training; instead, using as much data as possible to promote local performance is important.
-	If public datasets are available for pretraining, why don't we include them? It would be interesting to follow Nguyen et al.'s idea that uses some public data to pre-train the model and add this into comparison.
-	Why does the Non-IID setting present higher accuracy than the IID setting? When data becomes Non-IID, the model should suffer a performance drop.
-	Some Non-IID related methods (e.g., SCAFFOLD[1], FedDyn[2]) also aim to learn a good global model, which can serve as the initialized model. Adding them into comparison is helpful.
-	In Table 5, the centralized pretraining is based on a centralized dataset from all the clients; why does this introduce significant performance variance? If the centralized data are from all clients, the pretraining should present less variance than random initialization. Also, it needs to be clarified how to build the centralized dataset.

[1] Karimireddy, Sai Praneeth, Satyen Kale, Mehryar Mohri, Sashank Reddi, Sebastian Stich, and Ananda Theertha Suresh. "Scaffold: Stochastic controlled averaging for federated learning." ICML, 2020.

[2] Acar, Durmus Alp Emre, Yue Zhao, Ramon Matas Navarro, Matthew Mattina, Paul N. Whatmough, and Venkatesh Saligrama. "Federated learning based on dynamic regularization." ICLR,2021.

Overall, this paper studies an important topic, FL pretraining in a distributed manner, and also considers unseen data and fairness. The proposed method shows better performance and performance fairness than the compared methods. However, the experimental settings are not practical to me. Furthermore, the authors show a good motivation that some existing pretraining methods may suffer performance variances, but this interesting observation needs more in-depth analysis and explanations.

---

> ### Author Response · Authors · 2023-11-16
> **Response to Reviewer 4hah (1/4)**
>
> We appreciate the reviewer for the time and efforts, and providing constructive comments. Our responses are given below.
> ### **Comments regarding dataset splitting and unseen classes:**
> > We appreciate the reviewer for this comment. We would like to first make the following two points clearer:
> >
> > 1. In our setting, clients do not intentionally split their local dataset into pre-training dataset and downstream dataset: The clients participating during pre-training actually use all their local data samples to construct a pre-trained model that can adapt to any downstream FL tasks. This pre-training stage can be viewed as a “preparation phase” where clients use all their current data to *get prepared for various circumstances they may encounter in the future* during the downstream task. From the service provider’s perspective, as depicted in Fig. 1 of the manuscript, the pre-training stage can be viewed as a process to prepare a good product (i.e., a generalized initial model) that can be provided to any potential customers in the future (i.e., any group of clients in future downstream tasks).
> >
> > 2. The pre-training stage and the downstream stage share the same task domain (e.g., image classification for self-driving cars, disease predictions for patients), but their specific objective/task could be potentially different due to the **time-varying environment** or **new clients (that share the same task domain but with different objectives) joining the system**, e.g., classifying different objects depending on the self-driving car’s environment, classifying different diseases depending on the patient’s interest, or face/speech recognition for new phone users. Here, the clients participating during the downstream task could be either the one who participated during pre-training or the one who didn’t attend the pre-training stage but have become interested in the related task. For example, a self-driving car who participated during pre-training might become interested in another classification task depending on their location, weather condition, or other time-varying environments, which will make them to get interested in unseen classes. Here, even when clients observe unseen classes, they can still benefit from the pre-trained model that is pre-trained with the same task domain, rather than starting from a randomly initialized model. As another example, a new patient who didn’t participate during pre-training may want to take advantage of the pre-trained model for solving its own task with new diseases, instead of starting from a randomly initialized model.
> >
> > Under this scenario, our main question was: Can we design a pre-training methodology that can generalize well to any downstream FL tasks? Here, we would like to highlight that the downstream tasks are unknown during pre-training, and the downstream tasks may contain unseen classes, due to the time-varying scenarios and new clients joining the system as mentioned above. Note that if a specific downstream task is known during pre-training (i.e., if the set of classes for the downstream task is known), and if all those classes are available during pre-training, it is definitely possible to construct a specific pre-trained model tailored to that task by only utilizing the data samples that belong to those classes. However, this (i) requires the service provider to prepare different pre-trained models for every downstream task, which could be potentially infinite, and also (ii) requires the clients in the pre-training stage to contain all classes that will appear in any future downstream tasks, which is again impractical.
> >
> >&nbsp;
> >
> >Due to these challenges, given the current set of clients, the goal of the service provider would be to construct a “generalized pre-trained model” that can **quickly adapt to any, and possibly unseen future downstream tasks**. This is the goal of this paper, and we propose a meta-learning-based pre-training algorithm to mimic and generalize well to any unseen future downstream tasks. During the meta-learning process of our scheme, each client splits its local dataset into support set and query set to “mimic” downstream scenarios with potentially unseen classes, but we note that the clients are still taking advantage of all data samples for pre-training: The support sets are used to construct a temporary global model while the query sets are utilized for meta-update, which enables the clients to obtain a generalized initial model when pre-training is completed.   (to be continued in the next response)

---

> ### Author Response · Authors · 2023-11-16
> **Response to Reviewer 4hah (2/4)**
>
> > One might then ask what is the motivation of clients participating in pre-training. First, they can benefit from the pre-trained model for any of their future tasks that share the same task domain. Secondly, the service provider will provide incentives to the clients to promote their participation. The service provider is willing to do this since once a generalized pre-trained model is constructed, the service provider can sell/provide this pre-trained model to any group of clients or customers that share the same task domain. Thanks to the reviewer’s comment, we have made the above points clearer in our revised manuscript. Please let us know if the reviewer has any unclear points related to our response.
> >
> >&nbsp;
> >
> >To model the above practical scenarios with unseen classes during our experiments, we have split the dataset into 80 classes and 20 classes for pre-training and downstream, respectively. Note that this splitting is not done within the client, but naturally happens due to new clients joining the system and time-varying environments. However, as the reviewer pointed out, even in these scenarios, the classes observed during downstream FL might not always be the new, i.e., clients might still be interested in classifying some of the seen classes. To see the performance of different schemes under this scenario, we conducted additional experiments in which clients in the downstream FL task hold **'seen classes'**. To achieve this, we sampled 10 classes from the pre-training dataset and 10 classes from our CIFAR-100 downstream dataset, resulting in 20 classes with 10 seen classes and 10 unseen classes. Each time, we randomly selected 5 classes to conduct non-IID downstream FL, and we repeated the experiment 10 times. The pre-trained model was solely trained on the 80 classes of CIFAR-100 originally presented in our paper. The table below shows the results. Each FL pre-training method is trained on a non-IID data distribution and initializes the non-IID downstream FedAvg tasks using the pre-trained model. The results are consistent with the ones with only unseen classes, confirming the advantage of the proposed pre-training algorithm. These new results are now included in Section C.6 of Appendix.
> >
> > - Experiments with **mixed seen/unseen classes** during downstream FL:
> > |Pre-training method (Scenario I) |Acc | Var | worst 10% | worst 20% | worst 30%
> > |--|--|--|--|--|--|
> > | Centralized  | 82.63 | 63.57 | 67.35 | 69.22 | 70.38
> > |FedAvg | 80.19 | 51.35 | 68.72 | 70.15 | 72.33
> > | FedMeta | 83.14 | 39.85 | 67.29 | 71.35 | 73.81
> > | q-FFL  | 81.34 | 47.98 | 69.22 | 70.35 | 72.37
> > | CoPreFL | **84.79** | **30.51** | **70.83** | **72.66** | **74.1**
>
> ### **Side effects of centralized pre-training**
> >The centralized dataset is constructed by aggregating datasets from all clients during the pre-training phase. As evidenced in Tables 5, 38, 39, 40, and additional experiments provided below, centralized pre-training can introduce side effects such as performance biases and large variance. This phenomenon arises from the lack of generalizability in the model's design. When a model undergoes pre-training in a centralized manner based on SGD, it becomes rigidly bound to the knowledge in the pre-training dataset. This fixity poses a challenge in adapting the model to the diverse clients that may contain new or unseen data in downstream tasks, which arise due to the time-varying environment or new clients as described in our first response above. As mentioned, the key clarification is that the datasets used in the pre-training and downstream FL phases are disjoint, and the set of clients involved in these two phases is also not the same.
> >
> >&nbsp;
> >
> >Additionally, since our downstream tasks involve FL, the absence of learning dynamic distributed setups during the pre-training phase can further magnify this issue. Consequently, the model exhibits performance inconsistencies across different client datasets, leading to heightened performance variance. Our CoPreFL tackles these challenges by meta-updating the temporary global model to mimic FL scenarios that clients may encounter during downstream tasks. This enables our initial model to have generalization capability to arbitrary downstream FL scenarios. We have made these points clearer in Section I of the revised manuscript.

---

> ### Author Response · Authors · 2023-11-16
> **Response to Reviewer 4hah (3/4)**
>
> ### **Public  dataset**
> >Thank you for pointing this out. In response to the reviewer's suggestion, we have included an additional experiment considering a scenario where a public dataset is available. We conduct pre-training using the ImageNet_1K dataset following Nguyen et al.'s idea. We sample 200 images for each of the 1000 classes in ImageNet_1K. During the pre-training phase, we trained the model using the SGD optimizer with a learning rate of 1e-3 for 50 epochs, and considered this centralized training result as a baseline.
> >
> >&nbsp;
> >
> >Here, we would like to emphasize that, as stated in Remark 2 of our manuscript, even when provided with a public dataset, we can **intentionally split the dataset and apply our scheme** (using multiple computing units available at the server) to **mimic the distributed scenario of downstream federated learning and to capture generalization**, which can handle the limitation of the naive centralized pre-training as mentioned above. Hence, we also implemented our scheme using the ImageNet_1K dataset. For our method, we distribute all the data across 100 nodes, sampling 20 nodes in each round, and conducted our CoPreFL for 50 rounds. For the downstream FL phase, we apply non-IID FedAvg to the 20-classes downstream dataset we used in CIFAR-100 dataset. As a result, all classes observed during downstream tasks are the seen classes that have already appeared during pre-training.
> >
> >- Experiments with **public dataset** for pre-training (downstream on CIFAR-100):
> > |Dataset & Pre-training Method | Acc | Var|
> > |--|--|--|
> > |CIFAR100 pretrained – Centralized| 81.30 | 69.44 |
> > |CIFAR100 pretrained – FedAvg| 78.96 | 64.80 |
> > | CIFRA100 pretrained – FedMeta | 82.45 | 48.72
> > | CIFAR100 pretrained – qFFL | 80.01 | 88.92
> > | CIFAR100 pretrained – CoPreFL | 83.29 | 34.69
> > | ImageNet1K pretrained - Centralized | 86.75 | 67.34
> > | ImageNet1K pretrained - CoPreFL | **87.96** | **30.79**
> >
> > As we can expect, models pre-trained on ImageNet1K achieve higher accuracy compared to those pre-trained on CIFAR-100 since there is no unseen classes when pre-trained on ImageNet. More importantly, as mentioned, we can still apply our method by intentionally splitting the dataset and mimicking the distributed nature of downstream federated learning to achieve further performance improvements: The centrally pre-trained model on ImageNet achieves lower average performance and higher variance compared to our CoPreFL. This advantage of CoPreFL is achieved by initializing the model to get higher accuracy and better fairness in federated settings based on meta-learning. The overall results further confirm the advantage and applicability of our approach. These new results are now included in Section C.4 and Table 41 of Appendix.
>
> ### **Better results in Non-IID scenarios**
> >Intriguingly, we observed this notable phenomenon in our experiments, when applying initializations pre-trained on different data distribution, the non-IID downstream tasks exhibited higher accuracy than the IID setting in some cases. This results prompted us to delve deeper into the implications of using an initialization pre-trained on a diverse distribution. One plausible explanation for the higher accuracy in the non-IID setting could be attributed to the fact that pre-training in a non-IID manner allows the model to capture heterogeneity information within the data. This learned heterogeneity information may exhibit a certain degree of similarity with the distribution of the downstream tasks during the subsequent phases, thereby contributing to the improved performance observed in some non-IID scenarios. This finding raises interesting questions about the interplay between pre-training strategies and downstream task distributions, suggesting that non-IID pre-training might facilitate better adaptation to certain non-IID downstream tasks.

---

> ### Author Response · Authors · 2023-11-16
> **Response to Reviewer 4hah (4/4)**
>
> ### **Comparison with other FL method for pre-training**
> > Based on the reviewer's comment we  have conducted additional experiments by using FedDyn to pre-train the model on CIFAR-100. We consider IID and non-IID pre-training for FedDyn with $\alpha$ parameter of FedDyn is selected as 0.01. The downstream measurement is set to consist of 10 non-IID FedAvg tasks.The results show that our method still provides better initialization by taking both performance and variance into account during the pre-training phase. We have included these results in Section C.4 and Table 42 of Appendix.
> > - Experiments with other non-IID related method for pre-training :
> > |Pre-training method (Scenario I) |Acc | Var | worst 10% | worst 20% | worst 30%
> > |--|--|--|--|--|--|
> > | IID-FedDyn  | 83.15 | 40.13 | 69.55 | 71.32 | 73.95
> > | IID-CoPreFL | **84.36** | **38.56** | **73.66** | **75.63** | **77.40**
> > |--|--|--|--|--|--|
> > | NonIID-FedDyn  | 81.23 | 53.17 | 68.39 | 70.23 | 72.46
> > | NonIID-CoPreFL  | **83.29** | **34.69** | **71.58** | **73.20** | **74.59**
>
> Again, thank you for your time and efforts. Your comments are clear to the point and made us think carefully, and we feel we have managed to clarify all the issues raised: we tried to make our setting and goal clearer, justified why unseen classes can appear during downstream tasks with several examples, conducted additional experiments with mixed seen/unseen classes in downstream tasks, added new results using the public dataset, and provided further intuitions on centralized training and the performance on non-IID  scenarios. In case there are remaining questions/concerns, we hope to be able to have an opportunity to further answer them.

---

> > ### Comment · Reviewer_4hah · 2023-11-23
> >
> > Thanks for the detailed response. Some of my concerns have been checked. However, my major concerns have not been addressed.
> >
> > The authors did not reply to my second and fifth weakness points.
> >
> > **Comments regarding dataset splitting and unseen classes**: Regarding the problem setting, if the authors assume the pretraining stage and the downstream stage share the same task domain, then for a certain task, the scenario is more like "data-incremental" rather than domain or class incremental. For clients, it takes many efforts for a “preparation phase”. Their aim would be to maximally utilize the model for newly coming data. It is not common for a client to train a model and use it for some new tasks or some new unseen classes. References are needed to support the need of  ”time-varying environment” or “new “clients with different objectives”. Furthermore, it might be a bit misleading to use “pretraining” and “downstream” in this case.
> >
> > **Side effects of centralized pretraining**: I did not get why the centralized training rigidly bounds to the knowledge in the pretraining dataset. If the centralized training and FL training use the same dataset, why does this happen?

---

> ### Author Response · Authors · 2023-11-23
> **Further response to Reviewer 4hah**
>
> We appreciate the reviewer for the response. We apologize that we have missed some of your comments.
>
> &nbsp;
>
> ### **Weakness 2: Pre-training in non-IID setups**
>
> In our CoPreFL, each client’s local dataset does not need to necessarily represent the global distribution. As the reviewer understands correctly, our key idea is to use meta-learning to tackle downstream FL. Since the clients conduct meta-update on the global model using their own local dataset (which could be non-IID) and the server aggregates all the meta-updated models, our model is actually **learning to perform well across all  clients’ local datasets in the system**. This enables our pre-trained model to become a good initialization for downstream tasks; the constructed global model (starting from our initial model) performs well across all clients in the system during downstream. This is also confirmed via our extensive experimental results throughout the manuscript.
>
> &nbsp;
>
> ### **Weakness 5: Communication cost**
>
> Thanks for this comment. As mentioned in our response to the “public dataset above”, even when the public dataset is given, the central server can still adopt our CoPreFL to further enhance the performance (ImageNet1K - Centralized vs. ImageNet1K - CoPreFL).  Since the public dataset is centralized, CoPreFL can be also conducted at the server using multiple machines. 	The communication load between these machines in the server is the cost for achieving a better performance compared to centralized SGD.
>
> Conducting pre-training in a FL setup would become more important for the tasks in which the centralized public dataset is not large enough due to privacy issues. In healthcare applications, e.g., disease diagnosis, the available public datasets are not large enough to construct a strong pre-trained model that can generalize well, due to the privacy issues of collecting clients’ datasets. In such applications, different patients or hospitals can collaboratively pre-train the model based on their local datasets via FL, to construct a more robust, generalized initial model that can quickly adapt to any unseen downstream FL tasks. The communication cost in this process is a must  to construct a strong pre-trained model.
>
> &nbsp;
>
> ### **Comments regarding dataset splitting and unseen classes**
>
> We also appreciate this comment. We agree with the reviewer that from the client’s viewpoint,  the main goal would  be to prepare a model for current classes, rather than "future classes". Our problem setup is more important from the **service-provider’s perspective**. From the service provider's viewpoint, once a robust pre-trained model is constructed, the service provider can sell/provide this model as an initial model to **any group of clients that aims to construct a global model via FL**, in future downstream tasks (Fig. 1 in the manuscript). Our main question was: Can we construct this kind of robust pre-trained model using the currently available clients? The service provider will use the current clients as a tool to construct a pre-trained model, and will give incentives to the clients to participate in the pre-training stage.
>
> The service provider is willing to prepare this  especially considering  new clients joining the system (e.g., new self-driving cars joining a specific region [Badue et al]), which can introduce unseen classes.  Moreover, the clients’ needs  can also vary over time (e.g, as in continual FL settings [ICML’21], [CVPR’22]), and the clients may want to construct  a global FL model at a specific timestep.  These scenarios can be viewed as our key applications, and our work is suggesting the clients in these future downstream tasks to  conduct  FL  starting from our pre-trained model, instead of starting from a randomly initialized model or other pre-trained models.  **Overall, the ultimate goal of the service provider is to construct a strong pre-trained model that can adapt to any of these kinds of future downstream scenarios**. We have added the service provider's viewpoint in Section I of the manuscript.
>
>
> [Badue et al.,] ”Self-driving cars: A survey." Expert Systems with Applications 165 (2021): 113816.
>
> [ICML’21] Federated Continual Learning with Weighted Inter-client Transfer
>
> [CVPR’22] Federated Class-Incremental Learning
>
> &nbsp;
>
>
> ### **Side-effect of centralized pre-training**
>
> Simply conducting centralized training via SGD could potentially restrict the knowledge to the pre-training dataset, which degrades the performance when the dataset during downstream FL is different from the pre-training stage. Although our scheme uses the same dataset as centralized pre-training, the key difference of our CoPreFL is the use of  meta-learning. **Instead of simply biasing the knowledge to a specific dataset, CoPreFL learns how to adapt well to downstream FL scenarios** by meta-updating the global model, which leads to significant performance improvements.

---

> ### Author Response · Authors · 2023-11-23
> **Further response to Reviewer 4hah**
>
> ### **Weakness 4: Contribution over other meta-learning based methods**
>
> Some personalized FL works (e.g., FedMeta) also adopt meta-learning. We clearly differentiate our work with existing strategies in Sections I and II of our manuscript. Here, we provide a more detailed response.
>
>
>  The distinction between our CoPreFL and meta-learning based FL works (e.g., FedMeta) lies in the application, which requires us to conduct meta-update in a totally different way. FedMeta is designed to offer personalization to individual clients. In contrast, our emphasis is on creating an initial model that can construct a good “global model” during downstream instead of “personalized models”. This is the reason why we need to update the temporary global model instead of the local models, which is the key technical difference with FedMeta. Another technical difference is the linear combination between fairness and average performance term, as the reviewer understands correctly. These two key techniques enables CoPreFL to construct a robust initial model that can quickly adapt to “any group of clients” (instead of individual clients) to construct a global model during downstream tasks. The advantages of CoPreFL can be also confirmed via extensive experiments throughout the manuscript.
>
> Again, thank you for your time and efforts. We would appreciate further opportunities to answer any remaining concerns you might have.
>
>
> Best, Authors

---

### Official Review · Reviewer_khz5 · 2023-10-27

**Soundness:** 3 good
**Presentation:** 3 good
**Contribution:** 2 fair
**Rating:** 6
**Confidence:** 4

**Summary:**

This paper introduces CoPreFL, a pre-training approach for federated learning that aims at training a pre-trained model to serve as a good initialization point for FL. The authors demonstrated that by doing so, a better performing and fair global federated model can be achieved for federated clients. Specifically, the authors utilize meta-learning techniques and incorporate model performance variances among models as an additional term in the meta-objective to promote fairness. Interestingly, the method also additionally considers a scenario where the server holds a server dataset that can be used in the meta-training procedure. The paper also conducted empirical experiments to showcase the superior performance and fairness of the model trained by CoPreFL.

**Strengths:**

1. The paper is clearly written.
2. Handling of the scenario where the server holds a portion of the data is interesting and important, which could also be practical due to the wide presence of public datasets.
3. The experiments show promising results.

**Weaknesses:**

1. The paper lacks theoretical performance/fairness guarantees or analyses of the proposed algorithm.
2. The practicability of the method requires further justification, in terms of the number of tasks and amount of data required.

More details can be found below in the Questions section.

**Questions:**

1. Why is it the case that the existing works one federated meta learning for personalization are inapplicable to the problem setting of this paper? For the three papers (Jiang et al., 2019; Fallah et al., 2020; Collins et al., 2021) cited in the introduction, is it possible to simply replace the personalization stage of their algorithms with a federated global model training? If possible, can they be compared to as baselines in the experiment section?
2. In scenario II, why is it reasonable to use the server’s data as the unseen dataset for downstream tasks? How to make sure the artificial tasks generated from the server data are now a good representation of potential downstream tasks? The final goal of the algorithm is quick adaptation to the downstream tasks of clients, rather than the server. Is there a mismatch here for the optimization objective? Please justify this point since Scenario II is also an important claimed contribution of the paper.
3. The artificial partition of the server’s dataset into $|m|$ equal partitions also seems to be lacking justification. Also, empirically speaking, how well does your method work as compared to use (all local support data + all server data) for normal FedAvg training as the pre-trained model?
4. Can the authors briefly introduce FedMeta in the main text since it is an important baseline? Also, you probably need to highlight the difference between FedMeta and CoPreFL.
5. In reality, does the CoPreFL method require access to a large number of tasks to work well? For example in the experiments of this paper, we only perform 5-way classification for CIFAR-100 data, which comprises 100 classes. I am guessing that this is done so that a variety of tasks could be generated. Let’s say that my final goal is to train a good and fair model on CIFAR-100 now, how should I go about training the meta pre-trained model? Could the author clarify this point?
6. I believe it is a bit unclear in the paper that how does FedAvg and q-FFL (serving as baselines) do meta learning and hence can be compared to in the experiments.

---

> ### Author Response · Authors · 2023-11-19
> **Response to Reviewer khz5 (1/4)**
>
> We appreciate the reviewer for the time and efforts, and providing valuable comments. Our responses to the comments raised by the reviewer are given below.
>
> ### **Comparison with other federated meta-learning algorithms**
> > Yes, as the reviewer pointed out, it possible to replace the personalization stage (of existing federated meta learning methods) with a federated global model training and adopt as a baseline. We have done this for FedMeta in our original submission. Based on the reviewer’s comment, during the rebuttal period, we have further implemented another personalized FL method (Fallah et al.). The final global model from this approach is employed as an initialization in our downstream scheme. Specifically, we implement the personalized baseline under a non-IID pre-training data distribution and subsequently use the global model of this method to initialize the downstream non-IID FL tasks. The experiment is conducted on CIFAR-100 dataset, following the pre-training/downstream partition in our paper. The results are provided below:
> > - Comparison with modification of PerFedAvg (Fallah et al.):
> >|Pre-training Method (Scenario I) | Acc | Var | worst 10% | worst 20% | worst 30%
> >|--|--|--|--|--|--|
> >NonIID-PerFedAvg | 81.58 |49.73 |67.21|71.05|72.39
> >NonIID-CoPreFL | **83.29** | **34.69** | **71.58** | **73.20** | **74.59**
> >
> >The result further confirms the superiority of our method that meta-updates the temporary global model by balancing average performance and fairness. This table is now included in Section C.4 of the Appendix of the revised paper.
>
> ### **The utilization of server’s data to mimic unseen scenario, and partitioning server’s data**
>
> > We appreciate this comment. We would like to first highlight that we are not aiming to capture   representations of datasets in downstream FL. This is especially impossible when arbitrary and potentially new classes appear during downstream FL, which is the setup that we consider. In our setup, the pre-training stage and the downstream stage share the same task domain (e.g., image classification for self-driving cars, disease predictions for patients), but their specific objective/task could be potentially different due to the time-varying environment or new clients (that share the same task domain but with different objectives) joining the system, e.g., classifying different objects depending on the self-driving car’s environment, classifying different diseases depending on the patient’s interest, or face/speech recognition for new phone users. Here, the clients participating during the downstream task could be either the one who participated during pre-training or the one who didn’t attend the pre-training stage but have become interested in the related task. For example, a self-driving car who participated during pre-training might become interested in another classification task depending on their location, weather condition, or other time-varying environments, which will make them to get interested in unseen classes. Here, even when clients observe unseen classes, they can still benefit from the pre-trained model that is pre-trained with the same task domain, rather than starting from a randomly initialized model. To summarize, arbitrary “new classes” that have not appeared during pre-training may appear in downstream scenarios.
> >
> >&nbsp;
> >
> >Due to the above setting with arbitrary and potentially unseen classes, even the client’s local dataset may not be a good representation of the datasets of clients in downstream tasks. Meta-learning plays a key role here to tackle this challenge. Instead of directly capturing data representations for downstream, our initial model aims to learn “how to quickly adapt” to arbitrary unseen downstream FL scenarios, which is achieved by distinguishing between the support and query sets during pre-training. To facilitate this in Scenario II, we aimed to construct |m| different query sets based on the server’s data to “mimic downstream FL scenarios”, which is done by arbitrary partitioning. This is a natural choice since we are not able to know any patterns/information of downstream FL. What we have shown in Scenario II is that, even this arbitrary |m| partitioning of the server’s data can lead to significant performance improvements by mimicking a scenario that clients have different objectives/datasets in downstream scenarios. This can even improve the performance of the model compared to the one in Scenario I, as the clients can utilize all of its local dataset as the support in Scenario II.  These results further confirm the applicability and effectiveness of the proposed CoPreFL in Scenario II.

---

> > ### Author Response · Authors · 2023-11-19
> > **Response to Reviewer khz5 (2/4)**
> >
> > ### **Use (all local data + all server data) for normal FedAvg**
> > >We thank reviewer for the suggestion of comparing our pre-training scheme with FedAvg with all local data + all server data. The table below shows the results using CIFAR-100, confirming the advantage of CoPreFL.
> > > - Experiment compared with FedAvg with different data usage in scenario II (Pre-training: Non-IID; Downstream: Non-IID FedAvg):
> > >|Method | Acc | Var | worst 10% | worst 20% | worst 30%
> > >|--|--|--|--|--|--|
> > >FedAvg | 83.15 | 51.19 | 70.33 | 72.17 | 73.49
> > >CoPreFL | **86.63** | **31.58** | **73.05** | **75.82** | **77.58**
> >
> > ### **Difference compared with FedMeta**
> > >Thanks for reviewer’s suggestion. We now mention FedMeta in the related work section of the revised manuscript. We clarified the distinction between our method and FedMeta. A more detailed difference can be also found in the below answer:
> > >
> > >&nbsp;
> > >
> > > The distinction between our CoPreFL and FedMeta lies in the application, which required us to make the meta-update strategy different. Consider a healthcare application where each client, such as a hospital or an individual patient, aims to build a comprehensive global model capable of classifying a wide range of diseases. However, individual clients may possess limited types of diseases in their local datasets – for instance, one client may have data on diseases A and B but lacks information on diseases C and D. In this context, federated learning becomes essential even in the downstream tasks. Clients need to collaborate to construct a global model that not only reflects the diseases available locally but also incorporates information about diseases not present in their individual datasets, ensuring a more robust and universally applicable healthcare model. FedMeta is designed to offer personalization to individual clients, e.g., when a specific client is interested in predicting only diseases A and B. In contrast, our emphasis is on creating an initial model that can construct a good “global model” during downstream instead of “personalized models”, targeting the aforementioned applications. This made us to directly meta-update the temporary global model, which enables our CoPreFL to construct a robust initial model that can quickly adapt to “any group of clients” (instead of individual clients) to construct a global model during downstream tasks.

---

> ### Author Response · Authors · 2023-11-19
> **Response to Reviewer khz5 (3/4)**
>
> ### **Number of tasks during downstream in CoPreFL**
>
> > We appreciate the reviewer for pointing this out. Yes, we considered 5-way classification *during downstream FL to model various unseen downstream scenarios*. As mentioned in our second response above, our goal is to design a “generalized initial model” that can adapt to arbitrary downstream tasks that potentially contain unseen classes. To model unseen downstream scenarios, during experiments, we split the dataset into a pre-training dataset and a downstream dataset, allocating 80 and 20 classes for CIFAR-100 dataset, respectively. Here, **to prove that our scheme works well for “arbitrary” downstream scenarios, we had to generate different FL scenarios during downstream (i.e., by adopting 5-way classification) and measure the averaged performance over different downstream scenarios.** This allows us to assess whether the pre-trained model, generated using CoPreFL or other baselines, can offer effective initialization for a diverse range of downstream tasks. The goal is to evaluate the versatility of the pre-trained model in providing a robust starting point for various scenarios. However, this does not indicate that the number of downstream should be large in CoPreFL. To prove this, we now consider a “single downstream scenario“ with 20-way classification using all downstream classes (that have not appeared during pre-training). The results are provided below, indicating that CoPreFL still performs better than the baselines:
> > - Performance in a single-downstream scenario (Pre-training: Non-IID; Downstream: 20-way classification Non-IID FedAvg):
> > |Pre-training Method | Acc | Var | worst 10% | worst 20% | worst 30%
> >|--|--|--|--|--|--|
> >|FedAvg | 81.35 | 59.69| 70.35 | 70.91 | 71.63
> >|FedMeta| 82.69|46.38 |72.74|73.51|74.39
> >|q-FFL| 83.27|51.48|73.01|73.82|75.02
> >|CoPreFL | **84.71**|**39.26**|**73.28**|**74.99**|**75.35**
> >
> > Here, it is also important to mention that **CoPreFL targets scenarios where the target downstream tasks are “unknown” during pre-training**. If we have prior information on the downstream task (100-way CIFAR-100 classification) during pre-training, and if these classes are available during pre-training, there is no reason to design the initial model to perform well on arbitrary downstream tasks by adopting meta-learning. In such cases, it might be more beneficial to adopt simple FedAvg during pre-training to capture a good data representation for downstream FL, as pre-training and downstream tasks share the same classes. The strength of our CoPreFL lies in the robustness to capture arbitrary and potentially unseen downstream scenarios (as depicted in Fig. 1 of the manuscript) which happen due to the time-varying local datasets of clients or new clients (that share the same task domain but with different objectives) joining the system.
>
> ### **Regarding baselines FedAvg and q-FFL**
>
> >The comparison aims to evaluate the performance of downstream FL tasks initialized by pre-trained models trained using different FL methods. FedAvg and q-FFL do not involve a meta-learning design, so they perform their algorithms using selected clients during the pre-training phase. After obtaining the final global model for each method, we use it to initialize a fixed downstream FL task. We have included these schemes as a baseline since were not many baselines we could use for pre-training. We explored various federated learning algorithms (including those utilizing meta-learning differently from ours) and centralized training for pre-training baselines.

---

> > ### Author Response · Authors · 2023-11-19
> > **Response to Reviewer khz5 (4/4)**
> >
> > ### **Performance Guarantee**
> >
> > >We agree with the reviewer that a theoretical guarantee of the algorithm would make the paper stronger. We actually found this extremely challenging since it requires us to formally show “how the initial model constructed by each pre-trained model” affects the performance of the “global model constructed during downstream FL”. The authors of [1], which studies centralized pre-training for downstream FL, show that pre-trained model can help since it can potentially reduce the initial loss value in the beginning of downstream FL. However, when considering a general scenario involving arbitrary unseen downstream tasks, this might not always be the truth. Due to the challenge of providing exact theory, we aimed to consider as many baselines as possible to strengthen the empirical results. Since there were not many baselines we could use for pre-training, we explored various federated learning algorithms (including those utilizing meta-learning differently from ours) and centralized training for pre-training baselines. We evaluated the performance of these schemes over different data distribution setups for pre-training and downstream, and validated the superiority of our scheme. Nevertheless, we can still explain the superior performance of our scheme as follows: (i) We are adopting meta-learning, which is known to effectively handle unseen tasks, and (ii) we are taking advantage of meta-learning tailored to downstream FL by meta-updating the temporary global model considering both average performance and fairness, which enables us to construct a generalized initial model that leads to a strong global model. We believe that further investigating FL pertaining with theoretical analysis is a promising future research direction.
> > >
> > >&nbsp;
> > >
> > >[1] Where to Begin? On the Impact of Pre-Training and Initialization in Federated Learning, Nguyen et al.
> >
> >
> >
> >
> >
> > We again thank the reviewer for reviewing our paper and providing very helpful comments. We would appreciate further opportunities to answer any remaining concerns you might have.

---

> ### Comment · Reviewer_khz5 · 2023-11-23
> **Thanks for the responses**
>
> I would like to thank the authors for the detailed rebuttal, addressing my questions one by one. However, I am still not completely convinced by the usefulness of Scenario II, and the use of the server data by equal partitions in the proposed algorithm. Given that the paper is rather empirical in nature (without theoretical guarantees) while the experiments are rather weak due to the comparison to baselines (FedAvg and q-FFL) that are not designed for pre-training or meta-training, I cannot raise the score further for this paper.

---

> ### Author Response · Authors · 2023-11-23
> **Additional Response to Reviewer khz5**
>
> Thank you for your response and keeping the positive score. We would like to briefly make the point regarding equal partition clearer. Instead of equal partitioning, one can also divide the server-side dataset into partitions with unequal sizes and use them for  meta-update. However, during pre-training, the server does not know the dataset sizes of clients in future downstream tasks. In this case, one intuitive way is to treat all clients equally/fairly, by meta-updating the model with the same query set sizes. We show that this equal partitioning provides significant performance improvements as can be seen in our experiments.
>
> Regarding the baselines, there were not many baselines we could consider due to the lack of works on pre-training for/in FL, and thus we tried our best to include most of the variations we can think of (e.g., FedMeta, q-FFL, centralized pre-training). Thanks again for your time and efforts.
>
> Best, Authors

---

### Official Review · Reviewer_n2CX · 2023-11-02

**Soundness:** 2 fair
**Presentation:** 3 good
**Contribution:** 2 fair
**Rating:** 5
**Confidence:** 4

**Summary:**

Recent works showed that initializing federated learning (FL) with a pre-trained model can improve the FL performance compared to random initialization. However, all these works have considered the pre-trained model to be trained centralized and didn't tackle the scenario that the pre-training datasets are distributed across multiple clients and cannot be shared due to privacy issues. To fill in this gap, this paper exploited the idea of meta-learning and proposed an FL-based pre-training method to achieve a good initialization for unknown downstream FL tasks. Both the average performance and fairness across the FL participants are considered in the pre-training objective.

**Strengths:**

- Inspired by the meta-learning methods, this paper introduced a new FL pre-training method. Even though the idea of balancing between average performance and fairness is not new, it's the first time to be applied to the FL pre-training stage.

- The authors provided a relatively comprehensive literature review and clearly presented the novelty of the proposed method compared to existing works.

- The experimental results are impressive. The proposed algorithms are shown to outperform all the tested baselines in both average performance and fairness metrics.

**Weaknesses:**

My major concern is about the motivation of the proposed FL pre-training problem. I'm not convinced that an additional fairness-aware pre-training process is needed for FL. Specifically,

- This paper has focused on a setting where the pre-training dataset is not centralized. However, I wonder whether such a setting exists in any real-world application given so many open-sourced pre-trained models (e.g., Bert, GPT2, ViT, etc.) and large-scale public datasets (Common Crawl, Wikipedia, ImageNet, etc.) in different areas. Some real-world examples are needed to clarify the necessity of the FL pre-training stage.

- Even though the empirical results show that the proposed methods outperform the "Centralized" baseline, the centralized model tested in Table 5 is actually personally trained with a subset of data. There is no public large-scale pre-trained model tested in the experiments.

- In Section 4, FedAvg is used in the fine-tuning stage in all the experiments. However, I think the comparison between CoPreFL+FedAvg and Centralized+FedAvg is not enough to show the superiority of the proposed method. Why is the fairness issue must be tackled in the pre-training stage instead of the fine-tuning stage? How if I start from a public pre-trained model and do fine-tuning with any fairness-aware FL method (e.g., Centralized+FedProx)? Note that model pre-training is usually much more expensive than fine-tuning. Then, isn't it more sensible to consider the additional steps for optimizing (2) in the fine-tuning stage?

**Questions:**

-- In Eqs. (1) and (2), what does the $\mathcal{G}$ denote? Does $p(\mathcal{G})$ represent the distribution of a single downstream FL task or all the downstream tasks? If $p(\mathcal{G})$ is the distribution of a single task $\mathcal{G}$, there should be an additional expectation over $\mathcal{G}$ for both (1) and (2). Otherwise, the variance equation in (2) is weird. Why did you compute the variance of loss over clients in different FL tasks? For different downstream tasks, the loss function $f$ might be even not comparable.

-- In Algorithm 1 Line 17, How is the gradient computed on the server side with distributed data?

-- In Appendix B, the authors mentioned that they assessed various balancer values $\gamma$ in all scenarios and reported the best-performing value. How did you define the "best" here given the accuracy-fairness tradeoff?

---

> ### Author Response · Authors · 2023-11-16
> **Response to Reviewer n2CX (1/2)**
>
> We thank the reviewer for the comments and valuable feedback. Our responses to the comments raised by the reviewer are given below.
> ### **Key applications and the use of large-scale public dataset**
> > We appreciate the comment. Our responses can be summarized as follows:
> > 1. **Pre-training in a FL setup would be especially important for the tasks in which the centralized public dataset is not large enough due to privacy issues:** In healthcare applications, e.g., disease diagnosis, the available public datasets are not large enough to construct a strong pre-trained model that can generalize well, due to the privacy issues of collecting clients’ datasets. In such applications, different patients or hospitals can collaboratively pre-train the model based on their local datasets via FL, to construct a more robust, generalized initial model that can quickly adapt to any unseen downstream FL tasks.
> >
> > 2. **More importantly, as mentioned in Remark 2 of our manuscript, our scheme is directly applicable to centralized public datasets to further improve the performance:** As mentioned in Remark 2,  the server can intentionally split the centralized dataset and apply our scheme (using multiple computing units available at the server) to obtain a pre-trained model. Here the question is, what is the advantage of doing this compared to simple centralized training? As can be seen from Table 5, 38, 39, 40, centralized pre-training can introduce side effects such as performance biases and large variance. This phenomenon arises from the lack of generalizability in the model's design. When a model undergoes pre-training in a centralized manner based on SGD, it becomes rigidly bound to the knowledge in the pre-training dataset. This fixity poses a challenge in adapting the model to the diverse clients that may contain new or unseen data in downstream tasks, which arise due to the time-varying environment or new clients joining the system, e.g., classifying different objects depending on the self-driving car’s environment, classifying different diseases depending on the patient’s interest, or face/speech recognition for new phone users. Additionally, since our downstream tasks involve FL, the absence of learning dynamic distributed setups during the pre-training phase can further magnify this issue. Consequently, the model exhibits performance inconsistencies across different client datasets, leading to heightened performance variance. Our CoPreFL tackles these challenges by meta-updating the temporary global model to mimic FL scenarios that clients may encounter during downstream tasks. This enables our initial model to have generalization capability to arbitrary downstream FL scenarios. We have made these points clearer in Section I of the revised manuscript.
> >
> >To confirm the advantages of our CoPreFL over centralized pre-training in the large-scale public dataset, we have included an additional experiment considering a scenario where a public dataset is available. We conduct pre-training using the ImageNet_1K dataset following Nguyen et al.'s idea. We sample 200 images for each of the 1000 classes in ImageNet_1K. During the pre-training phase, we trained the model using the SGD optimizer with a learning rate of 1e-3 for 50 epochs, and considered this centralized training result as a baseline. For our method, we distribute all the data across 100 nodes, sampling 20 nodes in each round, and conducted non-IID CoPreFL for 50 rounds. For the downstream FL phase, we apply ten non-IID FedAvg tasks to the 20-classes downstream dataset we used in CIFAR-100 dataset. As a result, all classes observed during downstream tasks are the seen classes that have already appeared during pre-training.
> >
> >- Results using **public large-scale dataset** for pre-training (downstream on CIFAR-100):
> > |Dataset & Pre-training Method | Acc | Var|
> > |--|--|--|
> > |CIFAR100 – Centralized| 81.30 | 69.44 |
> > |CIFAR100 – FedAvg| 78.96 | 64.80 |
> > | CIFAR100 – FedMeta | 82.45 | 48.72
> > | CIFAR100 – qFFL | 80.01 | 88.92
> > | CIFAR100 – CoPreFL | 83.29 | 34.69
> > | ImageNet1K - Centralized | 86.75 | 67.34
> > | ImageNet1K - CoPreFL| **87.96** | **30.79**
> >
> >As we can expect, models pre-trained on ImageNet1K achieve higher accuracy compared to those pre-trained on CIFAR-100 since there is no unseen classes when pre-trained on ImageNet. More importantly, as mentioned, we can still apply our method by intentionally splitting the dataset and mimicking the distributed nature of downstream FL to achieve further performance improvements: The centrally pre-trained model on ImageNet achieves lower accuracy and higher variance compared to our CoPreFL. This advantage of CoPreFL is achieved by initializing the model to get higher accuracy and better fairness in federated settings based on meta-learning. The overall results further confirm the advantage and applicability of our approach. These new results are now included in Section C.4 and Table 41 of Appendix.

---

> > ### Comment · Reviewer_n2CX · 2023-11-23
> >
> > Thanks for your detailed responses. Can you clarify what pre-train model is used for ImageNet1K-Centralized? In Appendix C.4, you mentioned that "we initialized downstream FL using pre-trained ResNet weights (IMAGENET1K_V1) from the PyTorch library". However, in the response, it is said that the pre-trained model is centralized trained by the authors.

---

> ### Author Response · Authors · 2023-11-16
> **Response to Reviewer n2CX (2/2)**
>
> ### **Additional comparisons with other FL algorithm in fine-tuning stage**
>
> >We appreciate this comment. Based on the reviewer’s comment, we additionally consider FedProx, a more advanced FL algorithm that addresses heterogeneity compared to  FedAvg, to examine the robustness and generalizability of our pre-trained method. The results are provided in the table below. The additional experiments are conducted using the CIFAR-100 dataset. The FL methods in the pre-training phase are implemented under a non-IID data distribution, and we carry out 10 non-IID FedProx tasks for downstream evaluation. Here, we note that not only centralized pre-training but also other pre-training methods (including ours) can take advantage of FedProx during downstream tasks to further improve the performance. The results demonstrate that our pre-trained method maintains superiority in downstream FedProx compared to other pre-training methods. We would also like to note that the choice of FedAvg as our downstream task is made to minimize the varying impact introduced by other FL algorithms. Comparing the pre-training + downstream pairs, the improvement of CorPreFL+FedAvg over Centralized+FedProx shows that a better initialization, which considers the distributed scenario and balances fairness/performance in the pre-training phase, could potentially benefit the inferior downstream FL algorithm. These new results are now included in Appendix C.7.
> >
> > - Experiments with **applying FedProx in downstream FL**:
> > |Pre-training Method + Fien-tuning Method | Acc | Var | worst 10% | worst 20% | worst 30%
> > |--|--|--|--|--|--|
> > Centralized + FedProx  | 82.39 | 51.46 | 70.33 | 71.28 | 73.52
> > FedAvg + FedProx | 79.53 | 46.15 | 63.17 | 69.74 | 71.59
> > FedMeta + FedProx| 81.77 | 63.12 | 63.58 | 68.19 | 70.28
> > q-FFL + FedProx | 83.19 | 52.12 | 67.41 | 70.59 | 72.33
> > CoPreFL + FedProx | 84.31 | 30.55 |  70.19 |  73.88 | 75.13
> > CoPreFL + FedAvg | 83.29 | 34.69 | 71.58 | 73.20 | 74.59
>
>
> ### **Defination of $p(\mathcal{G})$**
> >In Eqs. (1) and (2), $p(\mathcal{G})$ represents the distribution of all possible sets of clients that appear in downstream. Hence, $\mathcal{G}$ is a random variable while G is a specific downstream task sampled from distribution $p(\mathcal{G})$. Based on these definitions, equations (1) and (2) can be also clearly defined without any issue.
>
>
> ### **Gradient computation in Algorithm 1 Line 17**
> >The gradient is computed on the client side. The communication process involves the server acquiring a temporary aggregated model from the local model trained using the support set (line 10). The server then broadcasts this model to each client for meta-evaluation, where gradients are computed locally on each client (lines 11-14). Subsequently, clients send these gradients back to the server, which utilizes them to update the temporary aggregated model (line 17). We have made these points clearer in Section 3.2 of the revised manuscript.
>
>
> ### **Best-performing value selection**
> >We report the results of our method, which achieved the best accuracy among all $\gamma$-configurations. Moreover, for fair selection/comparison, we also report the results of other baselines with their own best accuracy when searching for hyperparameters.
>
>
>
>
> Again, we appreciate the reviewer for the time and effort. Your concerns are clear, and we feel we have successfully addressed all the issues you raised. Please let us know in case there are remaining questions/concerns; we hope to be able to have an opportunity to further answer them.

---

> ### Author Response · Authors · 2023-11-23
> **Response to Reviewer n2CX**
>
> We appreciate your response. We pre-trained the model for “ImageNet1K - Centralized” by ourselves to make a *fair comparison with “ImageNet1K - CoPreFL” under the same setting*. The sentence in the Appendix needs to be removed. We have corrected it accordingly. Our goal here is to highlight that even when the public dataset is given, we can always apply our CoPreFL to mimic downstream FL scenarios, to construct a more robust initial model that can quickly adapt to arbitrary and possibly unseen downstream tasks. If a large public dataset is not available due to privacy issues (e.g., medical applications) as mentioned above, we must rely on pre-training in distributed settings.
>
> Please let us know if you have any further questions. Thank you.
>
> Best, Authors

---

### Official Review · Reviewer_8PDC · 2023-11-08

**Soundness:** 3 good
**Presentation:** 4 excellent
**Contribution:** 2 fair
**Rating:** 8
**Confidence:** 3

**Summary:**

This paper introduces CoPreFL, an approach that uses meta-learning techniques to create pre-trained models capable of adapting to various FL tasks while balancing utility performance and fairness across clients. The authors explore two scenarios: one where data is exclusively owned by clients and another where both clients and the server possess data. The paper provides empirical results on various settings using CIFAR100, TinyImageNet, and FEMNIST, along with comparisons to several FL baselines.

**Strengths:**

The paper is well-written, with clear and efficient communication of ideas. The federated learning settings and the problem addressed are highly relevant and important within the field of federated learning. The proposed solution exhibits good empirical performance on CIFAR100, TinyImageNet, and FEMNIST, in comparison to other baselines, even though the authors note that no baselines from other works are available for this particular task.

**Weaknesses:**

My main concern relates to the novelty of this approach and its performance guarantees. Let me elaborate further:

* **Novelty** - Algorithm 1 seems very similar to FedMeta [1] except for the objective which is a linear combination between a fairness term and a utility term.
* **Performance Guarantees** - CoPreFL currently lacks formal performance guarantees. This work would significantly benefit from a formal analysis of the proposed method(s) to provide a better understanding and confidence in their behaviour and their performance. Currently, evaluating the empirical results is challenging due to the nature of the baselines, which (to my understanding) have been adapted to accommodate pretraining and are not actual baselines (as the authors also identified) since they are optimized for entirely different scenarios.


[1] Federated Meta-Learning with Fast Convergence and Efficient Communication. Fei Chen, Mi Luo, Zhenhua Dong, Zhenguo Li, Xiuqiang He

**Questions:**

**Main questions:**

* What are the algorithmic differences between FedMeta and algorithm 1?

* Algorithm 2: What is the motivation behind applying the meta update to the temporary global model? Wouldn't applying the meta-update to the local models (akin to personalized FL) make more sense in the context of FL? Can you provide a real example of scenario 2?

* How should the support and query sets be selected in practice by each client?

 * What is the difference between the performance variance across clients during pretraining and downstream tasks for high $\gamma$ in the noniid settings?

* Experiments: Could you provide additional details regarding your selection process for the number of communication rounds and local epochs? Additionally, considering that you've reported performance on the worst-performing clients, is there a reason you didn't consider AFL or qFFL for larger values of q, such as q=20, which optimize for worst-case fairness across clients?

* Does the phrase "All of these schemes are adopted during the pre-training phase to construct initial models." in "Baselines for pre-training" mean that qFFL and FedAvg algorithms were modified to consider pertaining?

* Can you please explain CoPreFL performance on the Non-IID settings of tables 1, 2 and 3? How can data heterogeneity across clients have no large impact or even provide better results (especially given that local epochs are > 1)? My understanding is that the heterogeneity introduced across clients in the experiments is low.

*Also please see minor suggestions & typos:*


* section 2: "ptre-training stage"-> pre-training stage

* page 4 prior to Eq. 2: "achieving performance gains any unseen FL task"-> achieving performance gains on any unseen FL task

* Section 3.2.: the phrase "local training iterations" is misleading because the algorithm illustrates only a single local epoch, even though on experiments you consider 5.

* Fix the rephrasing in the abstract and conclusions to clarify that both scenarios are addressed: where clients own the data exclusively, and where both clients and the server own the data.

* page 5: "The objectives of our pre-trained model are to strike a balance" -> The objective of our pre-trained model is to strike a balance

---

> ### Author Response · Authors · 2023-11-18
> **Response to Reviewer 8PDC(1/3)**
>
> We thank the reviewer for the positive comments and valuable feedback. We appreciate the reviewer’s acknowledgment on the importance of our work. Our responses to the comments raised by the reviewer are given below.
> ### **Practical applications and differences with FedMeta**
> >**Key applications**: Consider a healthcare application where each client, such as a hospital or an individual patient, aims to build a comprehensive global model capable of classifying a wide range of diseases. However, individual clients may possess limited types of diseases in their local datasets – for instance, one client may have data on diseases A and B but lacks information on diseases C and D. In this context, federated learning becomes essential. Clients need to collaborate to construct a global model that not only reflects the diseases available locally but also incorporates information about diseases not present in their individual datasets, ensuring a more robust and universally applicable healthcare model. Similarly, in the domain of autonomous vehicles, each self-driving car may strive to develop a global model for scenario detection in various weather conditions. However, individual cars might encounter limited weather scenarios locally – one car might navigate through a desert environment, while another faces challenges in a snowy storm. Through federated learning, these cars can collectively construct a global model that accounts for a broad spectrum of weather conditions, ensuring robust scenario detection capabilities for all vehicles involved. The wide availability of datasets open to public also makes our Scenario II practical, as the server can hold these public datasets and meta-update the global model learned by clients.
> >
> >&nbsp;
> >
> > **Contributions beyond FedMeta**: The distinction between our CoPreFL and FedMeta lies in the application, which requires us to conduct meta-update in a totally different way.  FedMeta is designed to offer personalization to individual clients, e.g., when a specific client is interested in predicting only diseases A and B, or when a specific self-driving car is interested in the model tailored to a specific weather. In contrast, our emphasis is on creating an initial model that can construct a good “global model” during downstream instead of “personalized models”,  targeting the aforementioned applications. This is the reason why we need to update the temporary global model instead of the local models, which is the key technical difference with FedMeta. Another technical difference is the linear combination between fairness and average performance term, as the reviewer understands correctly. These two key techniques enables  CoPreFL to construct a robust initial model that can quickly adapt to “any group of clients” (instead of individual clients) to construct a global model during downstream tasks. The advantages of CoPreFL can be also confirmed via extensive experiments throughout the manuscript.
> >
> >&nbsp;
> >
> >We have clarified the distinction with FedMeta in the related work section of the revised manuscript.

---

> ### Author Response · Authors · 2023-11-18
> **Response to Reviewer 8PDC(2/3)**
>
> ### **Performance Guarantees and Baselines**
> >We agree with the reviewer that a formal analysis would make the paper stronger. We actually found this extremely challenging since it requires us to formally show “how the initial model constructed by each pre-trained model” affects the performance of the “global model constructed during downstream FL”. The authors of [1], which studies centralized pre-training for downstream FL, show that pre-trained model can help since it can potentially reduce the initial loss value in the beginning of downstream FL. However, when considering a general scenario involving arbitrary unseen downstream tasks, this might not always be the truth.  Due to the challenge of providing exact theory, we aimed to consider as many baselines as possible to strengthen the empirical results. Since there were not many baselines we could use for pre-training, we explored various federated learning algorithms (including those utilizing meta-learning differently from ours) and centralized training for pre-training baselines.  We evaluated the performance of these schemes over different data distribution setups for pre-training and downstream, and validated the superiority of our scheme. Nevertheless, we can still explain the superior performance of our scheme as follows: (i) We are adopting meta-learning, which is known to effectively handle unseen tasks, and (ii) we are taking advantage of meta-learning tailored to downstream FL by meta-updating the temporary global model considering both average performance and fairness, which enables us to construct a generalized initial model that leads to a strong global model. We believe that further  investigating FL pertaining with theoretical analysis is a promising future research direction.
> >
> >&nbsp;
> >
> > [1] Where to Begin? On the Impact of Pre-Training and Initialization in Federated Learning, Nguyen et al.
>
> ### **Parameter selection detail and q-value for qFFL**
> >For all FL pre-training methods, we consistently use 50 global rounds with each round of local training comprising 5 iterations per participant. This configuration has proven to be stable in our setup and is consistent with findings in prior work [1].  Regarding the choice of the q value in q-FFL, we selected values from the range [1, 3, 5] based on previous  research findings [2]. The authors have demonstrated that smaller q values (q > 0, i.e., 0.1 or 1) tend to result in better performance. On the contrary, higher q values might compromise performance and slow down the convergence speed. The selected range aimed to strike a balance between fairness considerations and optimization for better overall performance.
> >
> >&nbsp;
> >
> >[1] On the Importance and Applicability of Pre-Training for Federated Learning, Chen et al., ICLR ‘22.
> >
> >[2] Fair Resource Allocation in Federated Learning, Li et al., ICLR ‘20.
>
>
> ### **Better results in Non-IID scenarios**
> >Intriguingly, we observed this notable phenomenon in our experiments, when applying initializations pre-trained on different data distribution, the non-IID downstream tasks exhibited higher accuracy than the IID setting in some cases. This results prompted us to delve deeper into the implications of using an initialization pre-trained on a diverse distribution. One plausible explanation for the higher accuracy in the non-IID setting could be attributed to the fact that pre-training in a non-IID manner allows the model to capture heterogeneity information within the data. This learned heterogeneity information may exhibit a certain degree of similarity with the distribution of the downstream tasks during the subsequent phases, thereby contributing to the improved performance observed in some non-IID scenarios. This finding raises interesting questions about the interplay between pre-training strategies and downstream task distributions, suggesting that non-IID pre-training might facilitate better adaptation to certain non-IID downstream tasks.

---

> ### Author Response · Authors · 2023-11-18
> **Response to Reviewer 8PDC(3/3)**
>
> ### **Response for other questions**
> > - How should the support and query sets be selected in practice by each client?
> >>For each client, we randomly select 80% of their local data as support set and the remaining 20% as query.
> >
> >&nbsp;
> >
> > - Does the phrase "All of these schemes are adopted during the pre-training phase to construct initial models." in "Baselines for pre-training" mean that qFFL and FedAvg algorithms were modified to consider pertaining?
> >> Yes, the phrase means that we consider different baselines, including FedAvg  and q-FFL, during the pre-training phase using the pre-training dataset. The goal is to compare their ability to initialize the downstream FL tasks, which is the main focus of this work. We have included these schemes as baselines to strengthen the empirical results as there were not many baselines available for pre-training for FL.
> >
> >&nbsp;
> >
> > - What is the difference between the performance variance across clients during pretraining and downstream tasks for high gamma in the noniid  settings?
> >> Performance variance is assessed similarly during both pretraining and downstream tasks, involving the evaluation of the global model's performance on each participant's query/testing samples. However, the context in which variance is employed differs. During pretraining, performance variance serves as a factor to control the pretraining objective, utilizing a parameter, $\gamma$, to modulate its influence. A higher $\gamma$ encourages the pre-trained model to lean towards achieving better averaged accuracy performance, prioritizing accuracy over variance. Conversely, in downstream tasks, performance variance is merely a measure of the quality of the tasks and is independent of IIDness.
> >
> >&nbsp;
> >
> > - Other suggestions & typos
> >> All typos have been corrected in our revised paper.
>
>
> We again thank the reviewer for providing very helpful comments. We would love to answer any remaining questions the reviewer might have.

---

> > ### Comment · Reviewer_8PDC · 2023-11-23
> >
> > I thank the authors for their responses. I have revised my score accordingly.

---

> ### Author Response · Authors · 2023-11-23
> **Thank you for the reply**
>
> Thank you very much for acknowledging our efforts and raising your score.
>
> Best, Authors

---

### Author Response · Authors · 2023-11-20
**General Comments to AC and all Reviewers**

We appreciate the reviewers for their efforts and constructive suggestions, which have greatly helped us to improve the paper. Below, we would like to summarize how we have addressed the major comments raised by the reviewers.

**1. Utilizing large-scale public datasets (Reviewers n2CX and 4hah):**

In Section C.4 of the Appendix, we now include new experimental results with the ImageNet1k dataset for pre-training, and provide discussions.

**2. More detailed comparison with FedMeta (Reviewers 8PDC and khz5):**

In Section II, we now clearly introduce FedMeta and clarify its differences compared with our CoPreFL.

**3. More intuitions on the side effects of centralized pre-training (Reviewers n2CX and 4hah):**

In Section I, we elaborate on the side effects of centralized pre-training, strengthening our motivation.

**4. Our problem setting and key applications (Reviewers n2CX, khz5, and 4hah):**

 In Section C.8 of Appendix, we offer various real-world applications as examples to bridge the gap between theory and practice.  More detailed problem settings are also described in our response to each reviewer.

**5. Additional experiments to address reviewers' comments:**
- In Section C.6 of the Appendix, we consider both seen/unseen classes in downstream FL tasks (for Reviewer 4hah).
- In Section C.4 of the Appendix, we also incorporate two additional types of FL methods, non-IID related FL and personalized FL, as baselines in the pre-training phase (for Reviewers 4hah and khz5).
- In Section C.7 of the Appendix, we explore different FL algorithms as downstream tasks (for Reviewer n2CX).

For other details, please refer to our responses to each reviewer. Please let us know if you have any further comments. We would greatly appreciate the opportunity to answer them further.


Best, Authors

---

### Author Response · Authors · 2023-11-22
**General Comment to All Reviewers**

Dear Reviewers,

Thanks again for your time and efforts spent reviewing our paper. As mentioned in our response, we have carefully considered your comments and tried to address them, including **clarification on our problem setup and algorithm and additional experiments you expressed interest in**. We have also updated our manuscript accordingly. We are hoping to have an opportunity to hear your feedback and answer any other concerns you might have during the remaining Author-Reviewer discussion period, which ends within 34 hours.

Best, Authors of Paper 3574

---

### Meta-Review · Area_Chair_CTUf · 2023-12-09

**Metareview:**

This paper proposes a new approach to federated pre-training, carrying momentum from a recent line of work showing the importance of starting from a pre-trained model in federated learning. While previous work has focused on offline pre-training at the server, before starting federated training/fine-tuning, this work advocates for federated pre-training, arguing that it should be especially useful for applications where substantial public datasets are not available as in healthcare. The proposed approach seeks to balance between average performance across the population and fairness.

There was consensus that the paper is well-written and that the experiments reported in the paper are promising.

The major weaknesses raised were that the experiments are not sufficiently convincing because they rely on data which is not naturally organized into users, and hence it is not clear to what extent the results reported may carry over to federated applications of interest. The paper could be strengthened by demonstrating the improvements achieved on data from one of the domains used to motivate this work in the introduction. It was also unclear whether the primary focus of the paper is on federated pre-training or fairness in FL, and it would help to tease these two apart more clearly.

**Justification For Why Not Higher Score:**

Experiments are not sufficiently convincing of the relevance and significance of the work

**Justification For Why Not Lower Score:**

N/A

---

### Decision · Program_Chairs · 2024-01-16

Reject